# Towards Unified Human Motion-Language Understanding via Sparse Interpretable Characterization

**Guangtao Lyu[1], Chenghao Xu[1], Jiexi Yan[2]\*, Muli Yang[3], Cheng Deng[1]\***

[1] School of Electronic Engineering, Xidian University, Xi'an, Shaanxi, China,
[2] School of Computer Science and Technology, Xidian University, Xi'an, Shaanxi, China,
[3] Institute for Infocomm Research (I²R), A\*STAR, Singapore
{guangtaolyu,chx}@stu.xidian.edu.cn
{jxyan1995,muliyang.xd,chdeng.xd}@gmail.com

## Abstract

Recently, the comprehensive understanding of human motion has been a prominent area of research due to its critical importance in many fields. However, existing methods often prioritize specific downstream tasks and roughly align text and motion features within a CLIP-like framework. This results in a lack of rich semantic information which restricts a more profound comprehension of human motions, ultimately leading to unsatisfactory performance. Therefore, we propose a novel motion-language representation paradigm to enhance the interpretability of motion representations by constructing a universal motion-language space, where both motion and text features are concretely lexicalized, ensuring that each element of features carries specific semantic meaning. Specifically, we introduce a multi-phase strategy mainly comprising Lexical Bottlenecked Masked Language Modeling to enhance the language model's focus on high-entropy words crucial for motion semantics, Contrastive Masked Motion Modeling to strengthen motion feature extraction by capturing spatiotemporal dynamics directly from skeletal motion, Lexical Bottlenecked Masked Motion Modeling to enable the motion model to capture the underlying semantic features of motion for improved cross-modal understanding, and Lexical Contrastive Motion-Language Pretraining to align motion and text lexicon representations, thereby ensuring enhanced cross-modal coherence. Comprehensive analyses and extensive experiments across multiple public datasets demonstrate that our model achieves state-of-the-art performance across various tasks and scenarios.

## 1 Introduction

Understanding how we conceptualize and verbalize motion is a critical issue in the study of the human mind and communication. As a long-standing research hotspot, human motion understanding has led to the development of various tasks, including human motion-text retrieval (Petrovich et al., 2023; Yu et al., 2024), motion captioning (Guo et al., 2022b), and action recognition (Mahmood et al., 2019; Punnakkal et al., 2021), all of which have made significant progress in recent years.

Despite significant advancements in specific motion tasks, a considerable gap persists in achieving a nuanced comprehension of human motion, especially concerning fine-grained semantics and behavioral reasoning, since existing methods (Petrovich et al., 2023; Plappert et al., 2018; Zhou et al., 2024b) often lack interpretability in the context of motion understanding. For instance, as depicted in Fig. 1(a), when attempting to retrieve the motion associated with the phrase "Person is walking normally in a circle", the correct motion may be identified; however, the corresponding feature mapping within the unified cross-modal semantic space diverges significantly, failing to align with the semantic keywords essential for human comprehension. This semantic deficiency in the dense representations stems from their sole reliance on contrastive learning (Radford et al., 2021),

---

\*Corresponding author

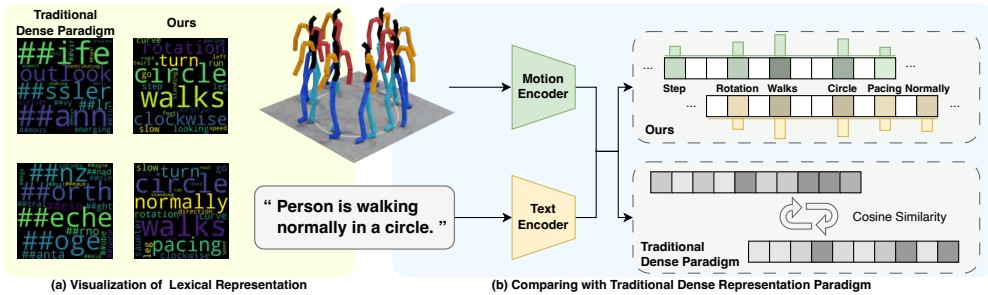

Figure 1: (a) Visualization of lexical representations of our framework and the traditional dense paradigm, and (b) conceptual comparison of our framework and the traditional dense paradigm. The color intensity reflects the higher values along the dimension.

which enforces cross-modal simply and rough alignment between dense text and motion embeddings primarily through cosine similarity, as illustrated in Fig. 1(b). Moreover, certain elements in text and motion are redundant and irrelevant. Such dense representations will incorporate this redundant information, which adversely affects the fine-grained semantic alignment of the representations. Consequently, it remains both crucial and challenging to further investigate interpretable representations of motion to bridge the understanding gap between language and human actions.

Lexical representation (Formal et al., 2021; Shen et al.; Xiao et al., 2022) integrates interpretability and efficiency by employing vocabulary elements to represent data, with each dimension corresponding to a specific term. The magnitude of each dimension indicates the significance of the associated term, enabling an intuitive understanding of data that aligns with human cognition, while the activation of only relevant vocabulary ensures sparsity and efficiency. In contrast, dense representation encapsulates data in a whole vector without clear dimensional significance, which limits their interpretability and obscures the underlying semantics. Drawing from this concept, we propose leveraging lexical representation to embed both motion and text within a cohesive lexical space, thereby enhancing clarity and fostering a more nuanced and intuitive comprehension of motion.

In this paper, we present a novel human motion-language pre-training framework that incorporates lexical representation to extract aligned sparse representations, thereby improving the interpretability of motion representations for better human motion understanding. Our method employs a multi-phase training strategy consisting of four key phases: i) Lexical Bottlenecked Masked Language Modeling (LexMLM), which enhances the pretrained language model's focus on high-entropy motion-related words for capturing the motion semantics; ii) Contrastive Masked Motion Modeling (CMMM), which improves motion feature extraction by directly capturing spatial and temporal dynamics from skeletal motion; iii) Lexical Bottlenecked Masked Motion Modeling (LexMMM), which enables the motion model to identify the underlying semantic features of motion, facilitating improved cross-modal understanding; and iv) Lexical Contrastive Motion-Language Pretraining (LexCMLP), which aligns motion and text representations within a unified vocabulary space to enhance cross-modal coherence. Through these training phases, our motion and language models effectively prioritize critical motion information and lexicon, yielding aligned and semantically rich sparse lexicon representations. Additionally, we conduct comprehensive visualization analyses of both motion and text across various modalities and perspectives. Extensive experiments on publicly available datasets demonstrate that our proposed method achieves state-of-the-art performance across diverse tasks and scenarios.

Our contributions can be summarized as follows:

- We integrate the lexical representation paradigm into the motion-language representation framework, aligning both motion and text within a shared lexical vocabulary space. This integration significantly enhances interpretability and fosters a more intuitive and comprehensive understanding of human motion.

- We propose a novel pre-training framework that learns aligned and semantically rich sparse lexical features for both motion and text, allowing the respective models to efficiently concentrate on key motion information and core vocabulary.

- We utilize advanced visualization techniques, including feature visualization and word clouds, to conduct thorough analyses of our method across motion, language, and cross-modal dimensions, thereby showcasing its interpretability and effectiveness.

- Comprehensive evaluations on several public datasets reveal that our model achieves state-of-the-art performance, affirming the efficacy of the proposed approach, while also showcasing its robust versatility across a wide range of tasks and scenarios.

## 2 RELATED WORK

**Human Motion Understanding.** In the motion-language domain, human motion understanding encompasses key tasks such as motion-text retrieval (Fujiwara et al., 2024b; Athanasiou et al., 2024; Fujiwara et al., 2024a) and motion captioning. **Motion-Text Retrieval** focuses on retrieving relevant motion data based on textual queries. A notable approach in this area is TMR (Petrovich et al., 2023; Bensabath et al., 2024), which constructs a cross-modal embedding space (Yan et al., 2024; 2021; Xu et al., 2024) through CLIP-style contrastive learning and negative filtering. MotionCLIP (Tevet et al., 2022a) further enhances this task by leveraging the image-text joint space. MotionPatch (Yu et al., 2024) utilizes motion patches along with pre-trained ViT weights for effective motion-language modeling. **Motion Captioning** aims to generate descriptive captions for human motions. TM2T (Guo et al., 2022b) encodes motion sequences and employs a translation network to bridge the gap between motion and text tokens. MotionGPT (Jiang et al., 2024) treats human motions as a foreign language, enabling descriptions through an expanded vocabulary.

**Lexical Representation.** Lexical representation is celebrated for its interpretability and efficiency, making it a popular choice in information retrieval (Formal et al., 2021). This approach highlights the importance of vocabulary in text representation and employs sparsity techniques and inverted indexing systems to enhance retrieval speed and reduce latency. Unlike traditional methods such as TF-IDF and BM25 (Robertson et al., 1995), which lack learning capabilities, neural-model-based lexical weighting retrieval methods (Shen et al.) leverage language models for term-centric searches. These methods fall into two main categories: those using causal language models (CLM)(Radford et al., 2018; 2019; Brown et al., 2020; Nogueira et al., 2019; Touvron et al., 2023; Zhang et al., 2022) and those based on masked language models (MLM)(Devlin, 2018; Liu, 2019; Sanh, 2019). However, a notable limitation is their tendency to fine-tune pre-trained language models directly, often overlooking the mismatch between general language modeling and relevance-driven lexical weighting (Luo et al., 2023; Zhou et al., 2024a; Chen et al., 2023). Moreover, the inherent semantic gap between motion and text presents significant challenges in aligning them within a shared embedding space. To address these challenges, we propose a novel multi-phase training framework specifically designed for motion-language learning, supported by a comprehensive analysis of its effectiveness.

## 3 METHOD

The primary objective of our methodology is to achieve seamless alignment of motion and text within a unified semantic space that captures the fundamental semantic features of motion, facilitating a comprehensive and in-depth understanding of motion. An illustration of our approach is shown in Fig. 2. We will discuss the critical components of the model architecture in Section 3.1 and the multi-phase training method in Section 3.2.

### 3.1 MODEL ARCHITECTURE

**Lexical Space.** We utilize the word embedding space of pre-trained language models (PLMs) space as our interpretable lexical space to encapsulate semantically meaningful relationships between motion and language, aligning seamlessly with human cognition. In contrast to conventional techniques like TF-IDF and BM25 (Robertson et al., 1995), this approach leverages advanced natural language processing methodologies (Devlin, 2018; Radford et al., 2018), including pre-training strategies and models developed on extensive language datasets, thereby significantly enhancing its capacity to capture the nuanced semantic information inherent in real-world human motions.

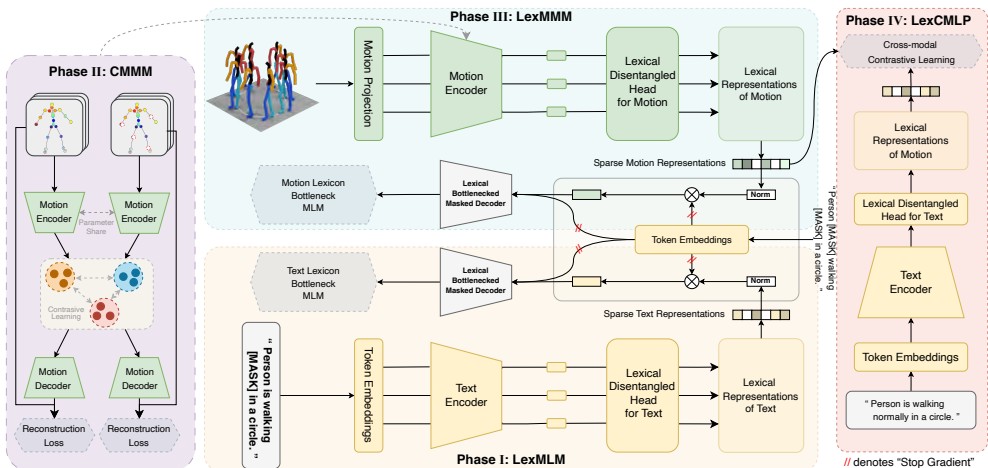

Figure 2: The framework of our method, including i) LexMLM, which enhances the language model's focus on high-entropy motion-related words; ii) CMMM, which captures spatial and temporal dynamics for robust motion representation; iii) LexMMM, which enables the motion model to identify semantic features and improve cross-modal understanding; and iv) LexCMLP, which aligns motion and text within a unified vocabulary space, ensuring cross-modal coherence.

**Text Encoder.** We utilize a pre-trained language model (Devlin, 2018) as our text encoder to effectively capture the semantic information from the input text. For an input motion caption of length $L$, denoted as $\mathbf{X} = [x_1, \ldots, x_L]$, the language model outputs text embedding with a shape of $\mathbb{R}^{L \times C_e}$, where $C_e$ represents the dimensionality of the common embedding space.

**Motion Encoder.** In contrast to existing methods (Jiang et al., 2024; Petrovich et al., 2023) that utilize specifically designed input formats and encoders for motion synthesis tasks Zhang et al. (2023); Petrovich et al. (2022); Guo et al. (2024); Athanasiou et al. (2022) in motion captioning and motion-text retrieval, our approach directly employs sequences of skeleton keypoints as inputs and designs a transformer equipped with both spatial and temporal attention to enhance interpretability. Formally, the input 3D motion sequence is represented as $\mathbf{X} \in \mathbb{R}^{T \times J \times C_{in}}$, where $T$ denotes the number of frames, $J$ indicates the number of joints, and $C_{in}$ specifies the input feature dimensions. We first project this sequence into a latent space $\mathbf{F}^0 \in \mathbb{R}^{T \times J \times C_f}$ and integrate learnable spatial joint position embeddings $\mathbf{P}_{pos}^S \in \mathbb{R}^{1 \times J \times C_f}$ along with temporal position embeddings $\mathbf{P}_{pos}^T \in \mathbb{R}^{T \times 1 \times C_f}$. The features are fed into the transformer to extract spatiotemporal information. Following feature extraction, a linear layer projects these features into the unified embedding space $\mathbf{E} \in \mathbb{R}^{T \times J \times C_e}$.

**Lexical Disentanglement Head.** The lexical disentanglement head is crafted to transform dense motion and text embeddings into sparse lexical representations. This process begins with the application of a language model head (LM-Head) (Devlin, 2018) space, which converts the motion and text from the embedding space into the lexicon space:

$$\boldsymbol{E}_{lex} = \text{LM-Head}(E_{embed}) \in \mathbb{R}^{|\mathcal{V}|}, \tag{1}$$

where $|\mathcal{V}|$ denotes the vocabulary size, $E_{embed}$ denotes the motion or text embedding and $\boldsymbol{E}_{lex}$ encapsulates the lexical representations corresponding to the input embeddings $E_{embed}$.

Subsequently, we adopt a strategy inspired by the SPLADE model (Formal et al., 2021) to reframe the data within a high-dimensional vocabulary framework:

$$p = \log(1 + \text{ReLU}(\text{MaxPool}(\boldsymbol{E}_{lex}))) \in \mathbb{R}^{|\mathcal{V}|}, \tag{2}$$

where the ReLU activation ensures non-negativity of all values and provides sparsity, MaxPool($\cdot$) conducts a maximum pooling operation along the sequence dimension and saturation function $\log(1 + \cdot)$ is utilized to prevent some terms from dominating the overall representation.

To further encourage sparsity, we follow (Paria et al., 2020) to introduce the FLOPs regularizer such that only a small number of token embeddings in $V$ are non-zeros:

$$\mathcal{R}_{\text{FLOPs}} = \sum_{k \in V} (\frac{1}{N} \sum_{i=1}^{N} p_k^{(i)})^2 \tag{3}$$

Consequently, the resulting vector $p$ serves as the sparse lexicon representation.

**Lexical Bottleneck Masked Decoder.** A key challenge in learning the lexicon-importance distribution lies in the absence of direct supervision. Moreover, PLMs (Devlin, 2018; Radford et al., 2018; 2019) are typically designed to reconstruct the entire text, which does not inherently prioritize motion-related key terms. To address this, we employ a Lexical Bottleneck Masked Decoder that reconstructs the entire text using dense representations derived from the sparse lexical representations. This approach enables the encoder to selectively emphasize motion-relevant tokens rather than treating all tokens with equal importance.

To facilitate this process, we propose a continuous bag-of-words (CBoW) (Shen et al.) derived from the lexicon-importance distribution $p$. Specifically, we define the bottleneck as follows:

$$E_{cbow} = p \cdot \text{sg}(W^{(te)}) \in \mathbb{R}^{|d|} \tag{4}$$

where $\mathbf{W}^{(te)} = [\mathbf{e}^{(w_1)}, \mathbf{e}^{(w_2)}, \ldots] \in \mathbb{R}^{|\mathcal{V}| \times d}$ is the learnable word embedding matrix from the language model, with $d$ indicating the embedding dim $C_e$ and $\mathbf{e}^{(w_i)} \in \mathbb{R}^d$ representing the word embedding for lexicon $w_i$, and $\text{sg}(\cdot)$ represents stop-gradient.

Finally, the bottleneck $E_{cbow}$ is fed into a simple decoder to reconstruct the masked tokens.

## 3.2 MULTI-PHASE LEXICAL-BOTTLENECKED PRETRAINING

The primary objective of our model is to integrate motion and text within a unified semantic space, accurately reflecting the intricate relationships between these modalities. However, significant challenges arise from the inherent differences between motion and language. Motion encoders often struggle to translate motion into lexical representations, while pre-trained language models (Devlin, 2018; Radford et al., 2018; Achiam et al., 2023), designed for sentence reconstruction, do not prioritize vocabulary relevant to motion. Additionally, relying solely on contrastive learning for modality alignment does not guarantee semantic accuracy. To tackle these challenges and ensure meaningful semantic correspondences, we propose an approach that goes beyond traditional contrastive learning. Our multi-phase pre-training strategy progressively aligns motion and text, preserving the semantic structure and emphasizing the importance of motion-related vocabulary.

### 3.2.1 LEXICAL BOTTLENECKED MASKED LANGUAGE MODELING

Pre-trained language models, typically trained with Masked Language Modeling (Devlin, 2018) or Next Token Prediction (Radford et al., 2018; 2019), focus on recovering words from context. Consequently, they often assign higher scores to low-entropy words, such as articles and prepositions, while neglecting high-entropy words crucial for distinguishing motion content. This highlights the necessity for additional pre-training to adapt PLMs for more effective lexical representation.

To address this challenge, we propose Lexical Bottleneck Masked Language Modeling (LexMLM), which consists of three key components: a Text Encoder, a Lexical Disentanglement Head, and a Lexical Bottleneck Masked Decoder. Following standard practices, we pre-train the text encoder using the MLM objective. Given a masked input $\bar{x}$, where 30% of the tokens are replaced by the [MASK] token, the text encoder generates embeddings and reconstructs the masked tokens. These embeddings are then converted into a sparse, meaningful lexicon representation $p$ by the Lexical Disentanglement Head, with sparsity enforced through a FLOPs regularizer. To further improve lexical representation and capture word importance without explicit labels, we propose the Bottleneck Masked Decoder to reconstruct the heavily masked input $\hat{x}$ using the learned lexicon representation $p$. We enhance this process with curriculum-masked modeling, gradually increasing the masking ratio from 50% to 100%. The whole LexMLM loss function is as follows:

$$\mathcal{L}_{\text{LexMLM}} = -\lambda_1 \sum_{\mathbb{D}} \sum_{j \in \mathbb{M}^{(\text{enc})}} \log P(\text{o}^j = x_j | \bar{x}) - \lambda_2 \sum_{\mathbb{D}} \sum_{j \in \mathbb{M}^{(\text{dec})}} \log P(\text{o}^j = x_j | \hat{x}; E_{cbow}) + \lambda_3 \mathcal{R}_{\text{FLOPs}}, \tag{5}$$

where $\mathbb{D}$ denotes the whole dataset, $\mathbb{M}^{(\text{enc})}$ and $\mathbb{M}^{(\text{dec})}$ denote the set of masked position in $\bar{x}$ and $\hat{x}$, $\text{o}^j$ denotes the logit of $x_j$, and $x_j$ refers to the original text token, the $E_{cbow}$ is calculated as Eq.4.

### 3.2.2 CONTRASTIVE MASKED MOTION MODELING

In the motion-language domain, much of the existing research (Guo et al., 2022b; Petrovich et al., 2023; Jiang et al., 2024) has focused on generation using handcrafted features, leaving a gap in the development of motion encoders that can directly capture both the spatial and temporal dynamics of skeletal motion for human motion understanding. To address this limitation, we propose a pretraining strategy for our motion encoder based on contrastive masked motion modeling (CMMM). In this approach, the motion sequence is processed through two parallel pathways: one with the full sequence $X$, and the other with a randomly masked portion of the input $X_m$. Both pathways share the same encoder weights, ensuring consistent feature extraction from both masked and unmasked inputs. To align the representations, we employ an InfoNCE loss (Oord et al., 2018) from contrastive learning, which encourages both pathways to produce similar embeddings, enhancing the encoder's ability to learn robust motion representations. Additionally, each pathway includes a simple decoder head that reconstructs the embeddings back into the motion sequence, $\hat{\mathbf{X}} \in \mathbb{R}^{T \times J \times C_{\text{in}}}$. This reconstruction process ensures that the learned features retain sufficient information to accurately recover the original sequence. By combining contrastive alignment with masked motion reconstruction, the encoder learns to capture robust motion features. The overall CMMM loss consists of reconstruction loss, velocity loss, and contrastive loss. The complete loss function is defined as:

$$\mathcal{L}_{\text{CMMM}} = \lambda_1 \sum_{t=1}^{T} \sum_{j=1}^{J} \| \hat{x}_{t,j} - x_{t,j} \|_2 + \lambda_2 \sum_{t=1}^{T-1} \sum_{j=1}^{J} \| \hat{v}_{t,j} - v_{t,j} \|_2 + \lambda_3 L_{info}(e_m, e_w), \quad (6)$$

where the $\hat{v}$ and $v$ mean the velocity of $\hat{x}$ and $x$, and $e_m$ and $e_w$ are the embedding of $x$ and $x_m$.

### 3.2.3 LEXICAL BOTTLENECKED MASKED MOTION MODELING

Despite the effectiveness of CMMM in extracting spatial-temporal motion embeddings, significant challenges persist in converting these embeddings into meaningful lexical representations. Unlike the extraction of lexical representations from text, this task requires a complex cross-modal conversion, where motion data must be translated into relevant textual representations. This process is inherently difficult due to the need to capture the intricate relationships between the dynamics of motion and the semantics of language, often resulting in the misalignment of representations.

To address this, we introduce Lexical Bottlenecked Masked Motion Modeling (LexMMM), composed of a Motion Encoder, a Lexical Disentanglement Head, and a Lexical Bottleneck Masked Decoder. The Motion Encoder extracts spatiotemporal motion embeddings, while the Lexical Disentanglement Head converts them into a sparse lexical representation, with a FLOPs regularizer applied to enforce sparsity. Similar to LexMLM, the Bottleneck Masked Decoder then reconstructs the masked text $\hat{X}$ input using the learned lexicon representation. We enhance the model with curriculum-masked modeling, gradually increasing the mask ratio from 50% to 100%. At full masking, the process can be interpreted as a motion captioning task. The whole LexMMM loss function is as follows:

$$\mathcal{L}_{\text{LexMMM}} = -\lambda_1 \sum_{\mathbb{D}} \sum_{j \in \mathbb{M}^{(\text{dec})}} \log P(\text{o}^j = x_j | \hat{x}; E_{cbow}) + \lambda_2 \mathcal{R}_{\text{FLOPs}}, \quad (7)$$

where $\mathbb{D}$ denotes the whole dataset, $\mathbb{M}^{(\text{dec})}$ denotes the set of masked position in $\hat{x}$, $\text{o}^j$ denotes the logit of $x_j$, and $x_j$ refers to the original text token, the $E_{cbow}$ is calculated as Eq.4.

### 3.2.4 LEXICAL CONTRASTIVE MOTION-LANGUAGE PRETRAINING

While the previous training stages effectively established a solid basis for generating semantic lexicon representations from both motion and text encoders, the independent tuning of these components resulted in suboptimal parameter configurations, thereby constraining the in-depth alignment between motion and text representations for comprehensive human motion understanding.

To achieve this, we introduce a novel cross-modal alignment strategy called Lexical Contrastive Motion-Language Pretraining (LexCMLP). This method utilizes motion and text encoder to form

| Setting | Methods | Text to motion retrieval | | | | | | Motion to text retrieval | | | | | |
|---------|---------|------|------|------|------|------|------|------|------|------|------|------|------|
| | | R@1↑ | R@2↑ | R@3↑ | R@5↑ | R@10↑ | MedR↓ | R@1↑ | R@2↑ | R@3↑ | R@5↑ | R@10↑ | MedR↓ |
| Dense | TEMOS | 2.12 | 4.09 | 5.87 | 8.26 | 13.52 | 173.0 | 3.86 | 4.54 | 6.94 | 9.38 | 14.00 | 183.25 |
| | T2M | 1.80 | 3.42 | 4.79 | 7.12 | 12.47 | 81.00 | 2.92 | 3.74 | 6.00 | 8.36 | 12.95 | 81.50 |
| | TMR | 8.92 | 12.04 | 16.33 | 22.06 | 33.37 | 25.00 | 9.44 | 11.84 | 16.90 | 22.92 | 32.21 | 26.00 |
| | MotionPatch | 10.80 | 14.98 | 20.00 | 26.72 | 38.02 | 19.00 | 11.25 | 13.86 | 19.98 | 26.86 | 37.40 | 20.50 |
| Lexicon | † TMR | 7.83 | 10.42 | 15.04 | 20.93 | 31.94 | 26.50 | 8.68 | 10.32 | 15.68 | 21.37 | 30.91 | 27.50 |
| | † MotionPatch | 9.13 | 12.86 | 16.78 | 23.83 | 34.71 | 22.50 | 10.03 | 11.89 | 17.13 | 23.44 | 33.38 | 24.50 |
| | Ours | **11.80** | **17.11** | **23.25** | **30.81** | **43.36** | **14.00** | **12.39** | **15.55** | **22.17** | **29.25** | **40.34** | **17.00** |

Table 1: Results on the motion-text retrieval benchmark on HumanML3D. The symbol † indicates that the lexicon representation is used directly in place of the dense embedding.

| Setting | Methods | Text to motion retrieval | | | | | | Motion to text retrieval | | | | | |
|---------|---------|------|------|------|------|------|------|------|------|------|------|------|------|
| | | R@1↑ | R@2↑ | R@3↑ | R@5↑ | R@10↑ | MedR↓ | R@1↑ | R@2↑ | R@3↑ | R@5↑ | R@10↑ | MedR↓ |
| Dense | TEMOS | 7.11 | 13.25 | 17.59 | 24.10 | 35.66 | 24.00 | 11.69 | 15.30 | 20.12 | 26.63 | 36.39 | 26.50 |
| | T2M | 3.37 | 6.99 | 10.84 | 16.87 | 27.71 | 28.00 | 4.94 | 6.51 | 10.72 | 16.14 | 25.30 | 28.50 |
| | TMR | 10.05 | 13.87 | 20.74 | 30.03 | 44.66 | 14.00 | 11.83 | 13.74 | 22.14 | 29.39 | 38.55 | 16.00 |
| | MotionPatch | 14.02 | 21.08 | 28.91 | 34.10 | 50.00 | 10.50 | 13.61 | 17.26 | 27.54 | 33.33 | 44.77 | 13.00 |
| Lexicon | † TMR | 9.87 | 12.13 | 19.64 | 28.19 | 42.16 | 15.50 | 10.62 | 11.18 | 20.07 | 27.13 | 36.51 | 18.00 |
| | † MotionPatch | 10.82 | 18.48 | 26.38 | 31.02 | 46.51 | 12.50 | 11.53 | 15.11 | 24.92 | 30.18 | 40.52 | 15.00 |
| | Ours | **15.13** | **23.74** | **31.61** | **36.81** | **54.12** | **8.00** | **15.01** | **19.47** | **30.06** | **35.63** | **47.53** | **10.50** |

Table 2: Results on the motion-text retrieval benchmark on KIT-ML. The symbol † indicates that the lexicon representation is used directly in place of the dense embedding.

contrastive pairs between sparse motion and caption lexicon representations. By applying a contrastive loss, we maximize the similarity between matching motion and text pairs while minimizing it for non-matching pairs. Additionally, to prevent semantic alignment in irrelevant spaces, the Lexical Bottleneck Masked Decoders from both LexMLM and LexMMM are retained as the lightly weighted auxiliary losses $L_{aux}^1$ and $L_{aux}^2$. This fine-tuning approach strengthens the alignment of lexical representations across modalities, ensuring that the motion and text lexicons are effectively synchronized within the shared vocabulary space. The whole LexCMLP loss function is as follows:

$$\mathcal{L}_{\text{LexMMM}} = \lambda_1 L_{info}(p_m, p_t) + \lambda_2 \mathcal{R}_{\text{FLOPs}} + \lambda_3 L_{aux}^1 + \lambda_4 L_{aux}^2, \tag{8}$$

where $p_m$ and $p_t$ represent the lexical representations of motion and text.

## 4 EXPERIMENTS

### 4.1 EXPERIMENTAL SETTINGS

**Datasets.** To validate the effectiveness of our sparse lexical representations, we conduct experiments on two commonly used public datasets: the HumanML3D (Guo et al., 2022a) dataset and the KIT Motion-Language dataset (Plappert et al., 2016). The HumanML3D dataset extends the AMASS (Mahmood et al., 2019) and HumanAct12 (Guo et al., 2020) motion capture datasets by adding natural language annotations, comprising 23,384 motions for training, 1,460 for validation, and 4,380 for testing. The KIT-ML dataset is focused primarily on locomotion, derived from motion capture data, 4,888 motions for training, 300 for validation, and 830 for testing.

**Evaluation Metrics.** We validate the representation alignment using the motion-text retrieval task and assess the semantics through the motion captioning task. To evaluate the alignment performance, we follow previous methods (Petrovich et al., 2023) and use standard metrics, including Recall at various ranks (R@1, R@5, etc.), and the median rank (MedR) for both text-to-motion (t2m) and motion-to-text (m2t) retrieval tasks. For evaluating semantic capability, we employ the same linguistic metrics as previous works (Jiang et al., 2024) including BLEU (Papineni et al., 2002), ROUGE (Lin, 2004), CIDEr (Vedantam et al., 2015), and BERTScore (Zhang et al., 2019).

**Implementation Details.** We utilize pretrained BERT (Devlin, 2018) as our text encoder and implement a transformer (Vaswani et al., 2017) with spatial and temporal attention mechanisms for the

| Methods | HumanML3D | | | | | KIT-ML | | | | |
|---------|-----------|-----------|--------|--------|-------------|----------|-----------|--------|--------|-------------|
| | Bleu@1↑ | Bleu@4↑ | Rouge↑ | Cider↑ | Bert Score↑ | Bleu@1↑ | Bleu@4↑ | Rouge↑ | Cider↑ | Bert Score↑ |
| TM2T | 48.9 | 7.00 | 38.1 | 16.8 | 32.2 | 35.1 | 6.2 | 28.7 | 28.9 | 30.4 |
| MotionGPT | 48.2 | 12.47 | 37.4 | 29.2 | 32.4 | - | - | - | - | - |
| Ours | **49.7** | **13.62** | **39.2** | **53.1** | **33.1** | **43.4** | **8.9** | **35.2** | **65.3** | **31.2** |

Table 3: Results on motion-to-text captioning benchmarks on HumanML3D and KIT-ML.

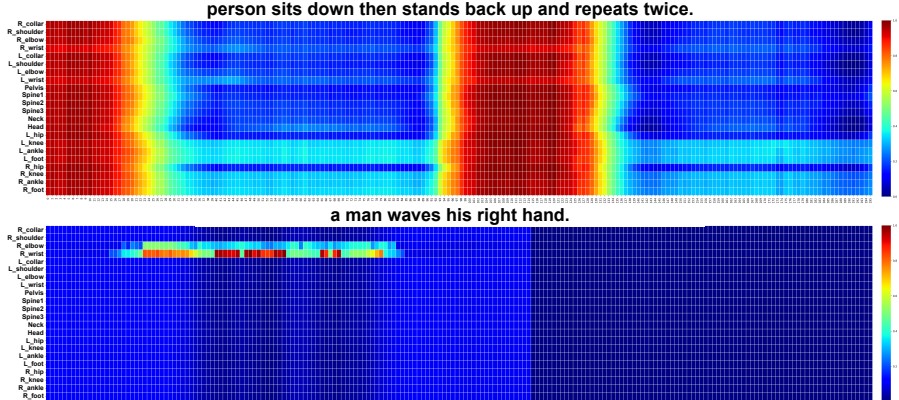

Figure 3: The PCA visualization of the spatiotemporal features extracted by our motion encoder.

motion encoder. Our experiments employ the Adam optimizer (Kingma & Ba, 2014), with learning rates set to $10^{-5}$ for the text encoder, $10^{-4}$ for the motion encoder, and $10^{-3}$ for the Lexical Disentanglement Head and Lexical Bottleneck Masked Decoder. During the LexMLM phase, we train with a batch size of 128 for 50 epochs. In the CMMM phase, we use a batch size of 64 and train for 200 epochs. For the LexMMM phase, we freeze the lexical space and fine-tune the motion encoder to align with the language domain, using a batch size of 64 for 150 epochs. Finally, in the LexCMLP phase, we use a batch size of 64 and train for 20 epochs at a learning rate of $10^{-5}$.

## 4.2 EXPERIMENTAL RESULTS

**Motion-Text Retrieval Results.** To evaluate the alignment between our text and motion lexical representations, we conduct motion-text retrieval benchmarks on the HumanML3D and KIT-ML datasets. We compare our proposed method with previous works, including TEMOS (Petrovich et al., 2022), T2M (Guo et al., 2022a), TMR (Petrovich et al., 2023), and MotionPatch (Yu et al., 2024). Additionally, we implement modified versions of TMR and MotionPatch by substituting their dense embeddings with lexicon representations. All evaluation metrics are consistent with those used in (Yu et al., 2024) to ensure a fair comparison. As illustrated in Tables 1 and 2, simply replacing dense embeddings with lexicon representations results in inferior performance, particularly for MotionPatch, indicating a significant loss of effectiveness from the pre-trained ViT (Dosovitskiy et al., 2020). In contrast, our method consistently outperforms existing approaches across all metrics in both datasets, demonstrating the effectiveness of our lexical representations and methods.

**Motion Captioning Results.** To evaluate the semantic capturing ability of our model, we conduct motion captioning benchmarks on the HumanML3D and KIT-ML datasets. We compare our approach with recent methods, including TM2T (Guo et al., 2022b) and MotionGPT (Jiang et al., 2024). As illustrated in Table 3, our method outperforms these contemporary techniques in generating text descriptions for specified motions. Notably, it excels in the CIDEr metric, which measures the model's effectiveness in capturing key information.

## 4.3 ABLATION STUDIES

**Ablation Studies on Multi-phase Training.** We conducted ablation studies on the KIT-ML to evaluate the effectiveness of our multi-phase pertaining approach. We implemented a CLIP-style baseline similar to prior works (Petrovich et al., 2023), incorporating a motion encoder, a text encoder, and contrastive loss. Additionally, we replaced the motion decoder with language decoder for motion captioning. We progressively introduced different training phases to elucidate their contributions. The

| Method | R@1 | R@5 | Cider |
|---|---|---|---|
| Baseline | 7.32 | 24.73 | 0.2 |
| + LexMLM | 8.82 | 26.17 | 0.8 |
| + CMMM | 12.98 | 32.73 | 0.9 |
| + LexMMM | 14.51 | 35.68 | 61.8 |
| + LexCMLP | 15.07 | 36.22 | 65.3 |

Table 4: Ablation experiments on different pre-training phases.

results reveal that the pre-training phases, CMMM and LexMLM, significantly enhance motion-text retrieval performance by improving the capabilities of both encoders. The LexMMM phase, which enriches the cross-modal understanding of the motion encoder, leads to notable gains in motion captioning metrics. The final alignment phase further elevates overall model performance. Detailed tuning for each phase is provided in the appendix.

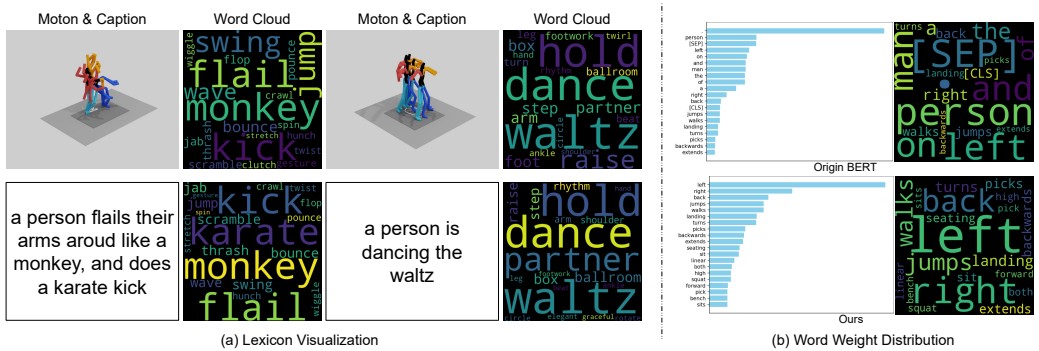

Figure 5: (a)Visualization of lexical representations of input motion and caption, and (b) the distribution of high-importance words extracted by the original BERT and ours.

**Ablation Studies on Sparsity.** Top-K Sparsifying (Shen et al.; Formal et al., 2021) adjusts the sparsity of lexicon-weighted representations, striking a balance between efficiency and effectiveness by retaining only the top-k weighted lexicons while setting others to zero. Applied exclusively during inference, this method introduces no additional training overhead. Fig. 4 illustrates the storage and retrieval performance across different sparsity levels on the KIT-ML dataset, where our model demonstrates superior storage efficiency and retrieval performance compared to previous approaches.

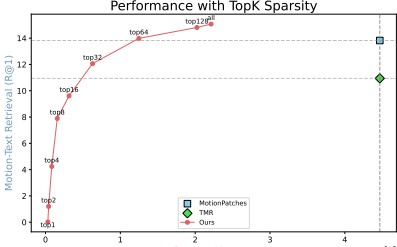

Figure 4: Results on KIT-ML dataset over different sparsity.

## 4.4 INTERPRETABLE QUALITATIVE ANALYSIS

**Lexical Representation Visualization.** We visualize the lexical representations of motion and text in Fig.5(a) using wordcloud(Oesper et al., 2011), where larger words represent higher lexical values. These visualizations show that the learned lexical representations align well with the motion semantics such as "monkey", "kick", "flail" and "dance waltz", highlighting the effectiveness of our method. Furthermore, the model accurately associates relevant words beyond those found in brief captions. For example, when inputting the "dance waltz" motion, our model learns classic movements like "hold" and related terms such as "partner", "ballroom" and "box step", reflecting its ability to efficiently explain the input motion by capturing comprehensive and contextually rich lexical information from both motion and text.

**Motion Features PCA Visualization.** We present the results of the principal component analysis (PCA) conducted on the motion spatial-temporal features extracted by our model. In this visualization, darker colors indicate stronger weights allocated by the model. As shown in Fig. 3, for the continuous local action "wave right hand", our model accurately maintains its focus on the right elbow and wrist. The subsequent frames are filled with blank spaces. In contrast, for the repetitive full-body motions "sit down" and "stand up", our model effectively captures this repetitive pattern and continuously attends to body parts such as the knee, ankle, and foot to a certain extent. This effective identification of relevant joints and dynamic emphasis aligns well with motion semantics, demonstrating that our model accurately extracts semantically relevant spatial-temporal motion features. Additional intuitive video demonstrations are provided in the supplementary materials.

**Word Weight Distribution Visualization.** Fig. 5(b) illustrates the frequency statistics of significant vocabulary before and after pre-training. Our text model demonstrates a stronger emphasis on motion-related vocabulary and relevant adjectives. In contrast, the original BERT (Devlin, 2018) tends to focus on low-entropy words such as "the", "of" and "a". The presence of terms like "person" and "man" highlights inconsistencies in motion caption styles within the dataset, including variations like "a person action", "the man action", "a figure action" and simply "action". As a result, the model also fixates on unnecessary details, such as punctuation variations in some sentences. Furthermore, the frequency distribution reveals imbalances, notably the significant disparity between

"left" and "right". These underscore potential quality issues in the dataset and provide insights for future analysis and improvements of the dataset and methods.

## 5 CONCLUSION

In this paper, we introduce the lexical representation paradigm to the motion-language domain, mapping motions and texts into a shared vocabulary space. We propose a novel multi-phase pre-training framework that efficiently learns aligned, semantically correct sparse lexicon representation for both modalities, which significantly enhances interpretability and depth in human motion understanding. Comprehensive analyses and extensive experiments on multiple public datasets, demonstrate that our model achieves state-of-the-art performance across various tasks and scenarios.

## 6 ACKNOWLEDGEMENTS

This work was supported in part by the National Key R&D Program of China (No. 2023YFC3305600), National Natural Science Foundation of China (62132016 and 62302372), Fundamental Research Funds for the Central Universities (ZDRC2102), and Natural Science Basic Research Program of Shaanxi (Program No. 2020JC-23).

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

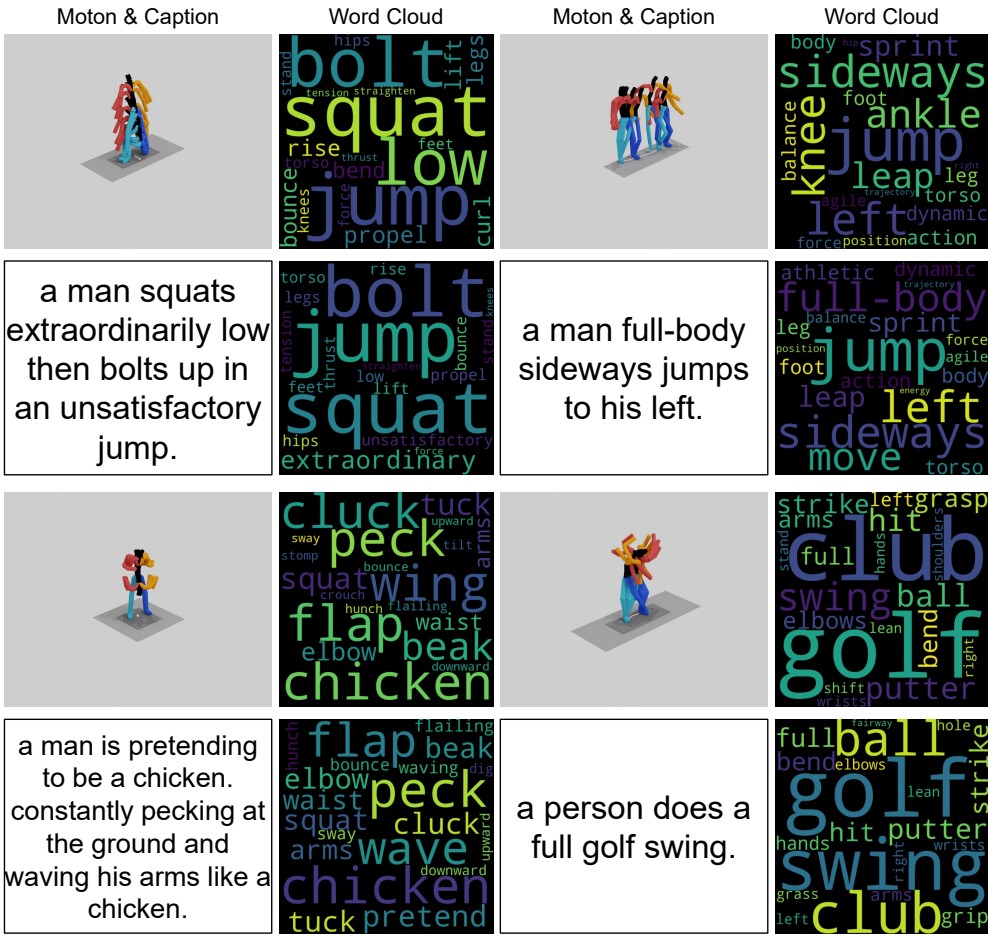

Figure 6: Visualization of lexical representations of input motion and caption.

## A  MORE RESULTS

### A.1  LEXICAL REPRESENTATION VISUALIZATION

We visualize the lexical representations of motion and text in Fig.6 using wordcloud, where larger words represent higher lexical values. These visualizations show that the learned lexical representations align well with the motion semantics such as "chicken", "jump", "swing" and "wave", highlighting the effectiveness of our method. Furthermore, the model accurately associates relevant words beyond those found in brief captions, reflecting its ability to efficiently explain the input motion by capturing comprehensive and rich lexical information from both motion and text.

### A.2  MOTION FEATURES PCA VISUALIZATION

We present the results of the principal component analysis (PCA) conducted on the motion spatial-temporal features extracted by our model in Fig.7. In this visualization, darker colors indicate stronger weights allocated by the model. Our model is able to focus on the correct body parts and track the motion changes along the temporal dimension. This effective identification of relevant joints and dynamic emphasis aligns well with the motion semantics, demonstrating that our model accurately captures semantically relevant spatial-temporal motion features.

### A.3  THE RESULTS OF TEXT TO MOTION RETRIEVAL

We present the results of text-to-motion retrieval in Fig.9. Our method effectively identifies keywords such as "monkey" and "scratch" retrieving results that are more closely aligned with the target motion. Although TMR and MotionPatches retrieve similar actions, the motions still exhibit noticeable differences compared to the "monkey" actions. In Fig.10, we further illustrate the results of retrieval using multiple keywords, which may better reflect typical search practices. Our method

**person is doing a hand stand.**

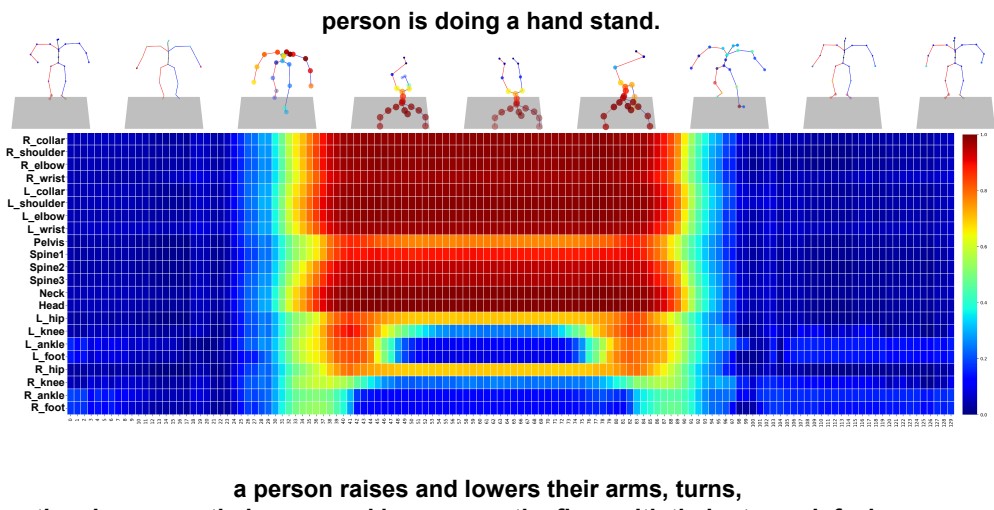

**a person raises and lowers their arms, turns,
then lowers so their arms and legs are on the floor with their stomach facing up.**

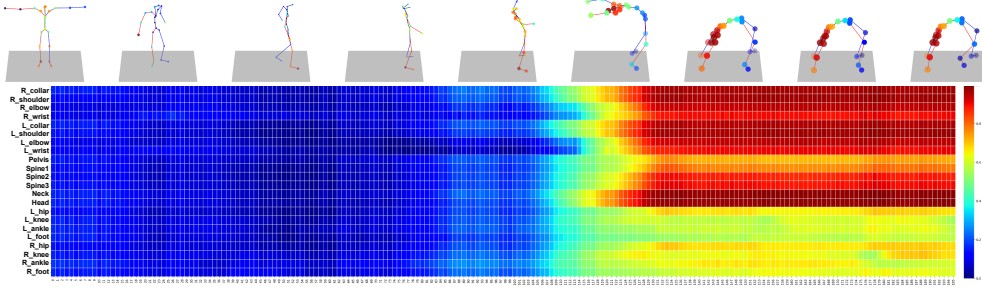

**a person is walking in place at a slow pace.**

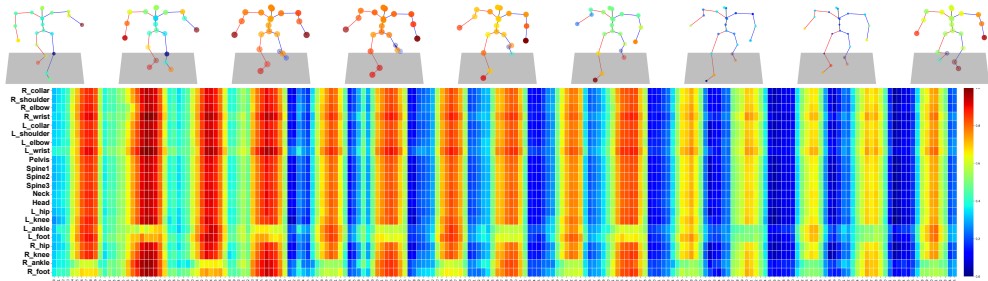

Figure 7: The PCA visualization of the spatiotemporal features extracted by our motion encoder.

ensures that the retrieved motions are simultaneously related to both "run" and "jump" while existing methods tend to miss one of the keywords. These results demonstrate the effectiveness of our approach, especially in handling keyword-based retrieval. In addition, our model also supports non-parametric indexing, that is, without using a text encoder, the weight of the keyword is directly set to 1. It can also be combined with techniques such as sparse storage and inverted indexing to further improve the efficiency and effectiveness of retrieval.

## A.4 THE RESULTS OF MOTION CAPTIONING

We present the results of motion captioning in Fig.8. The captions generated by our method exhibit greater distinction, capturing more detailed and relevant keywords. For instance, in the dataset, swimming is a common action, but our approach effectively identifies specific details like "butterfly stroke" and "standing". Similarly, for waltz, in addition to the general term, our method highlights unique movements such as the "box step". This showcases the model's ability to capture critical information, further demonstrating the effectiveness of our approach.

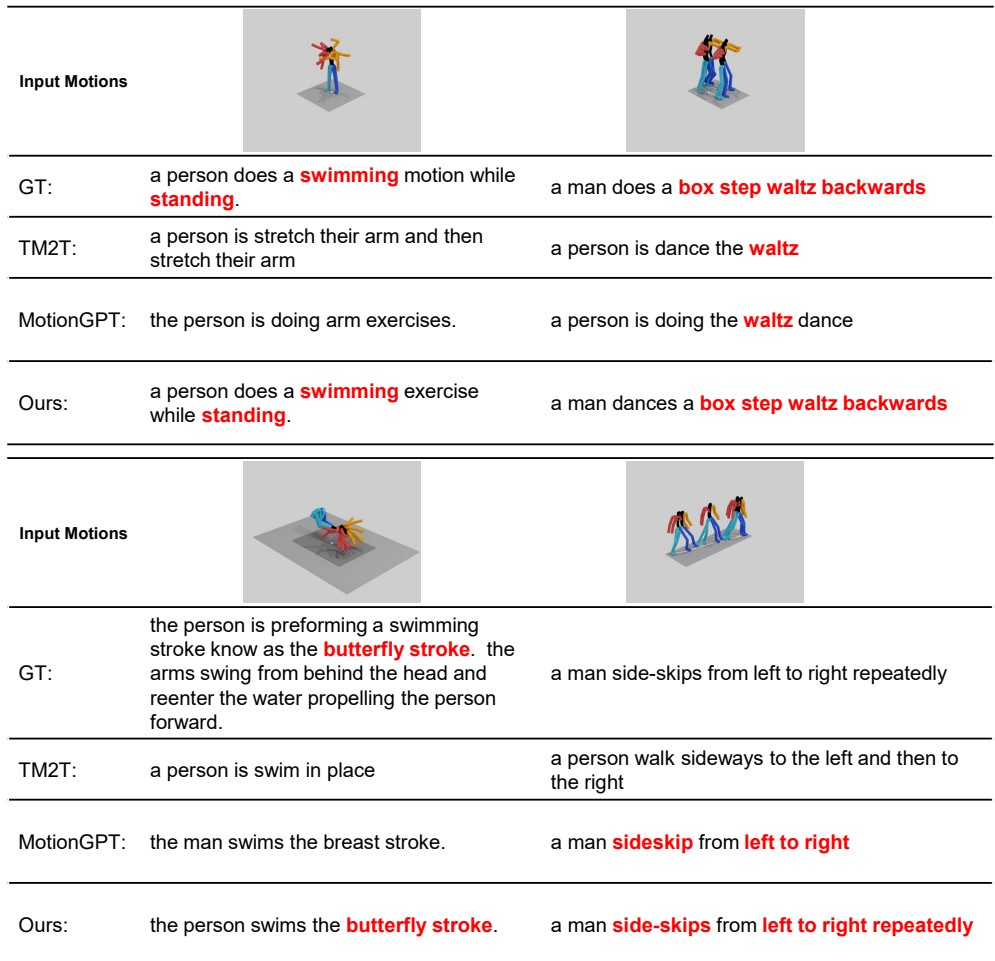

Figure 8: The motion captioning results. The red words highlight the keywords.

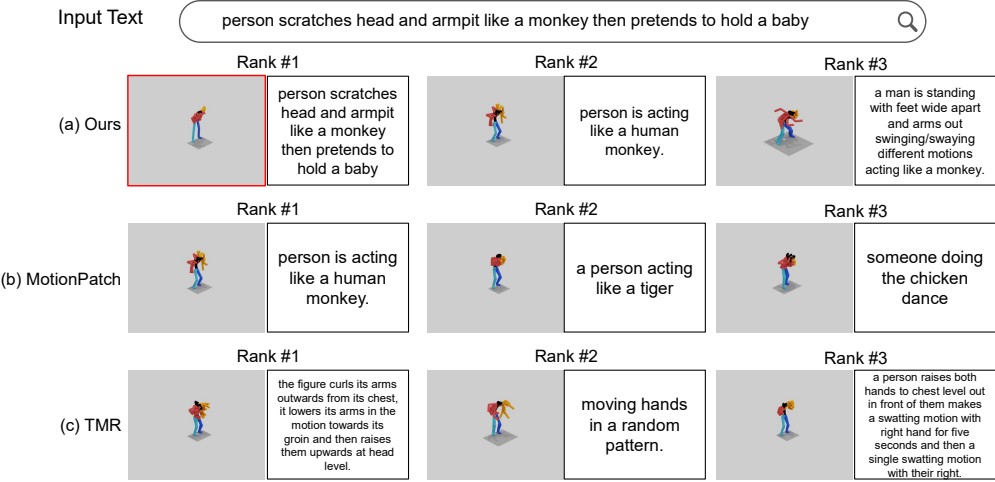

Figure 9: The text to motion retrieval results. The red box means the right sample.

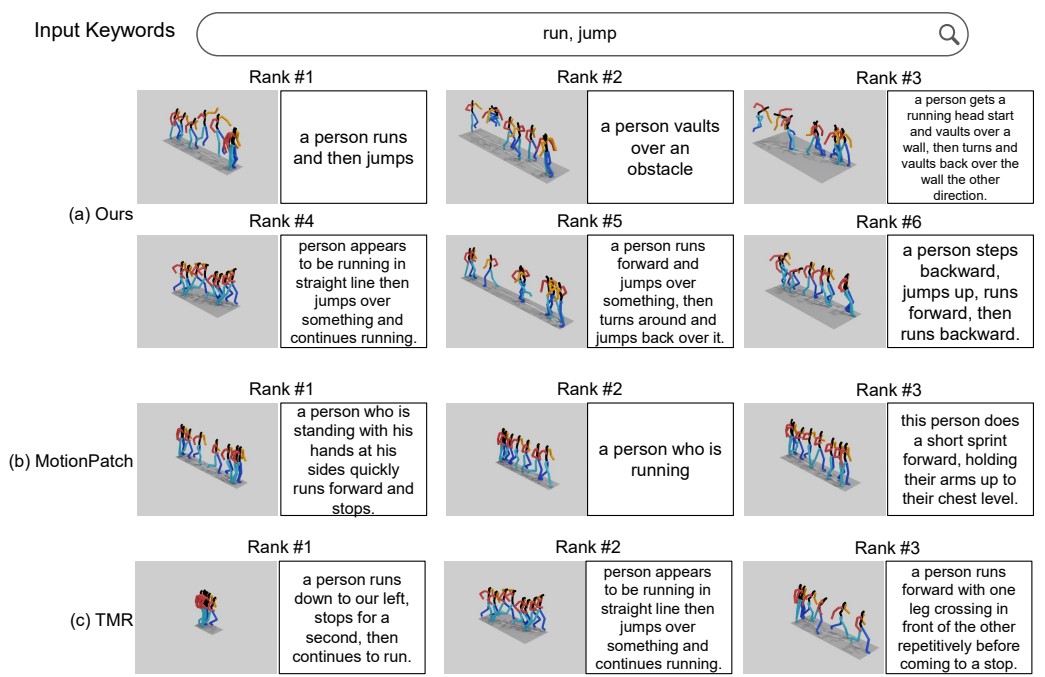

Figure 10: The keywords to motion retrieval results.

# B    DETAILED ABLATION STUDIES

## B.1    LEXICAL BOTTLENECKED MASKED LANGUAGE MODELING

As shown in Table 5, we conducted ablation experiments on the hyperparameters of LexMLM loss using the KIT-ML dataset. The results highlight the importance of MLM pretraining for the text encoder and the significance of pretraining the Lexical Bottleneck Masked Decoder. Furthermore, assigning a larger weight to $\lambda_2$ improves performance, demonstrating that lexical bottlenecked masked modeling is more effective for extracting key terms. Overall, the ablation study illustrates the effectiveness of the different components in LexMLM.

| $\lambda_1$ | $\lambda_2$ | $\lambda_3$ | R@1 | R@5 | Cider |
|---|---|---|---|---|---|
| 1 | 0 | 1e-4 | 7.91 | 24.87 | 0.2 |
| 0 | 1 | 1e-4 | 7.62 | 24.79 | 0.2 |
| 1 | 1 | 1e-4 | 8.13 | 25.31 | 0.4 |
| 0.1 | 1 | 1e-4 | 8.37 | 25.81 | 0.5 |
| 0.1 | 1 | 1e-5 | 8.31 | 25.66 | 0.5 |

Table 5: Ablation experiments of the hyperparameters of LexMLM on KIT-ML.

As shown in Table 6, we conducted ablation experiments on different mask ratios of the LexMLM loss using the KIT-ML dataset. Increasing the mask ratio in the encoder yields better results, while a higher mask ratio in the decoder further improves performance. This reflects the strength of both standard Masked Language Modeling and our Lexical Bottleneck Masked Language Modeling approach. However, directly setting the mask ratio to 100% harms the model's effectiveness. To address this, we adopted a curriculum learning strategy, gradually increasing the mask ratio from 50% to 100%. These results demonstrate the impact of varying mask ratios and the effectiveness of our method.

| $\mathbb{M}^{(enc)}$ | $\mathbb{M}^{(dec)}$ | R@1 | R@5 | Cider |
|---|---|---|---|---|
| 15% | 15% | 8.31 | 25.66 | 0.5 |
| 15% | 30% | 8.35 | 25.71 | 0.5 |
| 30% | 30% | 8.42 | 25.79 | 0.5 |
| 30% | 50% | 8.55 | 25.88 | 0.6 |
| 50% | 50% | 8.31 | 25.69 | 0.6 |
| 30% | 70% | 8.73 | 25.95 | 0.7 |
| 30% | 100% | 8.38 | 25.76 | 0.5 |
| 30% | 50-100% | 8.82 | 26.17 | 0.8 |

Table 6: Ablation experiments of the mask proportion of LexMLM on KIT-ML.

## B.2    CONTRASTIVE MASKED MOTION MODELING

As shown in Table 7, we conducted ablation experiments on different mask ratios and the weights of the contrastive loss in CMMM using the KIT-ML dataset. For the reconstruction loss, we used both MSE loss and velocity loss, each with a weight of 1. Our primary focus is on demonstrating the impact of different mask ratios and contrastive loss weights. The results indicate that increasing the mask ratio appropriately improves performance. Moreover, incorporating the contrastive loss further enhances the retrieval capability of the model. These ab-

| mask | $\lambda_3$ | R@1 | R@5 | Cider |
|------|-------------|-------|-------|-------|
| 0% | 0 | 10.86 | 29.17 | 0.8 |
| 15% | 0 | 11.48 | 29.93 | 0.8 |
| 30% | 0 | 11.73 | 30.53 | 0.8 |
| 50% | 0 | 11.97 | 31.02 | 0.8 |
| 60% | 0 | 11.86 | 30.83 | 0.8 |
| 70% | 0 | 11.81 | 30.81 | 0.8 |
| 50% | 1 | 12.62 | 32.03 | 0.9 |
| 50% | 0.1 | 12.98 | 32.73 | 0.9 |
| 50% | 0.01 | 12.58 | 32.26 | 0.9 |

Table 7: Ablation experiments of CMMM on KIT-ML.

lation studies highlight the effectiveness of both masked motion modeling and contrastive motion modeling, validating the overall performance of CMMM. of both masked motion modeling and contrastive motion modeling, validating the overall performance of CMMM.

### B.3 Lexical Bottlenecked Masked Motion Modeling

As shown in Table 8, we conducted ablation experiments on different mask ratios in LexMMM using the KIT-ML dataset. Increasing the mask ratio appropriately enhances the model's capability. We also applied curriculum learning, gradually raising the mask ratio from 40% to 100%. Through cross-modal pretraining with LexMMM, our method significantly improves the model's semantic understanding, helping it extract the correct keywords. This leads to improved retrieval performance. Overall, the experiments demonstrate the effectiveness of our LexMMM pretraining, particularly in enhancing semantic understanding.

| $\mathbb{M}^{(dec)}$ | R@1 | R@5 | Cider |
|-----------------------|-------|-------|-------|
| 15% | 13.85 | 34.71 | 49.5 |
| 30% | 14.02 | 35.13 | 54.7 |
| 40% | 14.36 | 35.39 | 57.3 |
| 50% | 14.27 | 35.31 | 57.9 |
| 60% | 14.12 | 35.15 | 57.1 |
| 70% | 13.92 | 34.95 | 56.8 |
| 100% | 13.57 | 34.12 | 51.9 |
| 40-100% | 14.51 | 35.68 | 61.8 |

Table 8: Ablation experiments of the mask proportion of LexMMM on KIT-ML.

### B.4 Lexical Contrastive Motion-Language Pretraining

As shown in Table 9, we conducted ablation experiments on different hyperparameters in Lex-CMLP using the KIT-ML dataset. A smaller weight was assigned to preserve the MLM loss obtained from the Lexical Bottleneck Masked Decoder, forcing the model to output accurate semantic information, which enhances its understanding and representational capacity. The experimental results demonstrate that our Lex-

| $\lambda_1$ | $\lambda_2$ | $\lambda_3$ | $\lambda_4$ | R@1 | R@5 | Cider |
|-------------|-------------|-------------|-------------|-------|-------|-------|
| 1 | 1e-4 | 1e-2 | 0 | 14.62 | 35.79 | 62.4 |
| 1 | 1e-4 | 0 | 1e-2 | 14.82 | 35.96 | 63.7 |
| 1 | 1e-4 | 1e-2 | 1e-2 | 15.07 | 36.22 | 65.3 |
| 1 | 1e-4 | 1e-3 | 1e-3 | 15.01 | 36.13 | 65.1 |

Table 9: Ablation experiments of the hyperparameters of LexCMLP on KIT-ML.

CMLP further improves the alignment of lexical representations and the model's ability to accurately grasp semantic information, proving the effectiveness of our method.

| Text Encoder | Parameters | Text to motion retrieval | | | | | | Motion to text retrieval | | | | | |
|---|---|---|---|---|---|---|---|---|---|---|---|---|---|
| | | R@1↑ | R@2↑ | R@3↑ | R@5↑ | R@10↑ | MedR↓ | R@1↑ | R@2↑ | R@3↑ | R@5↑ | R@10↑ | MedR↓ |
| T5-Small | 80M | 4.40 | 7.14 | 10.02 | 14.19 | 21.25 | 60.00 | 5.66 | 6.66 | 9.88 | 13.85 | 19.79 | 68.00 |
| T5-Base | 250M | 4.95 | 7.62 | 10.31 | 14.81 | 23.62 | 46.00 | 5.59 | 6.98 | 10.59 | 14.49 | 20.99 | 54.00 |
| T5-Large | 780M | 5.82 | 8.72 | 11.66 | 16.93 | 26.27 | 37.00 | 6.69 | 8.33 | 12.16 | 16.70 | 23.53 | 45.00 |
| T5-XL | 3B | 7.39 | 11.00 | 14.65 | 20.65 | 31.29 | 28.00 | 7.94 | 10.06 | 15.08 | 19.88 | 29.05 | 33.00 |
| T5-XXL | 11B | 8.41 | 12.82 | 15.96 | 23.67 | 34.42 | 24.00 | 8.97 | 11.49 | 16.69 | 22.63 | 32.16 | 28.00 |
| DistilBERT | 66M | 10.80 | 14.98 | 20.00 | 26.72 | 38.02 | 19.00 | 11.25 | 13.86 | 19.98 | 26.86 | 37.40 | 20.50 |

Table 10: Ablation Study of the Motivation on the Motion-Text Retrieval Benchmark Using the HumanML3D Dataset. While performance improves with the increasing size of T5 models, the largest T5-XXL still underperforms compared to the smaller DistilBERT.

## C    MORE TABLES AND FIGURES

### C.1    MORE ABLATION STUDIES ABOUT THE MOTIVATION

To further clarify the motivation behind our method, we conducted an ablation study on the the existing state-of-the-art approach, MotionPatch Yu et al. (2024), using a series of text encoders of varying model sizes, ranging from T5-Small to T5-XXL Chung et al. (2024); Raffel et al. (2020). The experimental results are summarized in Table 10. It is evident that increasing the model size of T5 leads to improved performance. However, despite its significantly larger size, the performance of the T5 model remains substantially inferior to that of the original text encoder, DistilBERT Sanh (2019). This indicates that employing a more powerful text encoder does not necessarily enhance the alignment of semantic keywords critical for human understanding. These observations are supported by prior studies. For example, in the ablation study of MotionPatch Yu et al. (2024), a more advanced text encoder failed to deliver improved representational performance. Similarly, TMR Petrovich et al. (2023) advised the use of DistilBERT instead of the larger CLIP series text encoders. Additionally, TMR++ Bensabath et al. (2024) analyzed text annotations across various datasets (HumanML3D, KIT-ML, Babel) and demonstrated that while text augmentations can partially bridge the domain gap, significant differences persist between datasets. Hence, simply adopting a more powerful text encoder does not address the core issue identified in our study. A deeper and more nuanced understanding of the correlation between textual descriptions and motion is required. Consequently, the design of a lexical representation in our paper is both necessary and justified.

### C.2    EXTENSION TO TEXT-TO-MOTION GENRATION TASK

Our method can be seamlessly integrated into text-to-motion (T2M) generation tasks. Currently, by simply replacing the text encoder in existing methods with our proposed text encoder to extract a more refined and sparse lexical representation, we observe consistent performance improvements across various generative architectures. Comparative experiments were conducted on multiple frameworks, including VAE-based methods (e.g., T2M Guo et al. (2022a)), diffusion-based approaches (e.g., MDM Tevet et al. (2022b)), autoregressive models (e.g., T2M-GPT Zhang et al. (2023)), and non-autoregressive models (e.g., MoMask Guo et al. (2024)). The results, summarized in Table 11, show notable gains, particularly in metrics evaluating motion-text alignment, such as R-precision (Top-1, Top-2, Top-3 accuracy) and Multimodal Distance (MM-Dist). These improvements are primarily attributed to two factors: (1) CLIP-based text encoders, pre-trained on image-text datasets, do not sufficiently focus on motion-related content, and (2) these encoders fail to capture temporal information during the text-image pretraining process. In contrast, our text encoder is specifically designed to identify motion-relevant keywords during pretraining and align them with a motion encoder that captures spatiotemporal motion information, leading to enhanced performance in T2M tasks. This aligns with recent observations in specialized T2M studies, such as HumanTomato Lu et al. and LGTM Sun et al. (2024), which highlight similar limitations of original CLIP text encoders. However, the majority of mainstream T2M research still emphasizes improving generation paradigms to achieve better results.

As shown in Fig. 11, when multiple actions are present in a sentence, the original MoMask often overlooks some actions, resulting in poor performance. This improvement arises because the pretrained CLIP-text encoder does not prioritize motion-related information, and the lack of temporal information in images makes it less sensitive to the sequential nature of actions. In contrast, ours

| Paradigms | Methods | FID ↓ | Top1 ↑ | Top2 ↑ | Top3 ↑ | MM-Dist ↓ |
|---|---|---|---|---|---|---|
| VAE | T2M | $1.087^{\pm.021}$ | $0.455^{\pm.003}$ | $0.636^{\pm.003}$ | $0.736^{\pm.002}$ | $3.347^{\pm.008}$ |
| | T2M$^{\S}$ | $0.942^{\pm.009}$ | $0.472^{\pm.004}$ | $0.653^{\pm.002}$ | $0.748^{\pm.003}$ | $3.104^{\pm.006}$ |
| Diffusion | MDM | $0.544^{\pm.044}$ | $0.320^{\pm.005}$ | $0.498^{\pm.004}$ | $0.611^{\pm.007}$ | $5.566^{\pm.027}$ |
| | MDM$^{\S}$ | $0.524^{\pm.036}$ | $0.357^{\pm.004}$ | $0.536^{\pm.003}$ | $0.643^{\pm.005}$ | $5.212^{\pm.021}$ |
| AR | T2M-GPT | $0.141^{\pm.005}$ | $0.492^{\pm.003}$ | $0.679^{\pm.002}$ | $0.775^{\pm.002}$ | $3.121^{\pm.009}$ |
| | T2M-GPT$^{\S}$ | $0.133^{\pm.005}$ | $0.506^{\pm.004}$ | $0.684^{\pm.003}$ | $0.781^{\pm.004}$ | $3.002^{\pm.006}$ |
| NAR | MoMask | $0.045^{\pm.002}$ | $0.521^{\pm.002}$ | $0.713^{\pm.002}$ | $0.807^{\pm.002}$ | $2.958^{\pm.008}$ |
| | MoMask$^{\S}$ | $\mathbf{0.041}^{\pm.002}$ | $\mathbf{0.532}^{\pm.002}$ | $\mathbf{0.721}^{\pm.003}$ | $\mathbf{0.814}^{\pm.002}$ | $\mathbf{2.852}^{\pm.008}$ |
| VAE | T2M | $3.022^{\pm.107}$ | $0.361^{\pm.005}$ | $0.559^{\pm.007}$ | $0.681^{\pm.007}$ | $3.488^{\pm.028}$ |
| | T2M$^{\S}$ | $2.836^{\pm.062}$ | $0.372^{\pm.004}$ | $0.574^{\pm.004}$ | $0.695^{\pm.005}$ | $3.235^{\pm.016}$ |
| Diffusion | MDM | $0.497^{\pm.021}$ | $0.164^{\pm.004}$ | $0.291^{\pm.004}$ | $0.396^{\pm.004}$ | $9.191^{\pm.022}$ |
| | MDM$^{\S}$ | $0.482^{\pm.009}$ | $0.214^{\pm.005}$ | $0.319^{\pm.004}$ | $0.418^{\pm.005}$ | $8.682^{\pm.014}$ |
| AR | T2M-GPT | $0.514^{\pm.029}$ | $0.416^{\pm.006}$ | $0.627^{\pm.006}$ | $0.745^{\pm.006}$ | $3.007^{\pm.023}$ |
| | T2M-GPT$^{\S}$ | $0.502^{\pm.016}$ | $0.423^{\pm.005}$ | $0.641^{\pm.006}$ | $0.752^{\pm.006}$ | $2.927^{\pm.015}$ |
| NAR | MoMask | $0.204^{\pm.011}$ | $0.433^{\pm.007}$ | $0.656^{\pm.005}$ | $0.781^{\pm.005}$ | $2.779^{\pm.022}$ |
| | MoMask$^{\S}$ | $\mathbf{0.186}^{\pm.009}$ | $\mathbf{0.441}^{\pm.006}$ | $\mathbf{0.668}^{\pm.004}$ | $\mathbf{0.792}^{\pm.005}$ | $\mathbf{2.693}^{\pm.013}$ |

Table 11: Evaluation on the Text-to-Motion Generation Benchmarks: HumanML3D dataset (upper section) and KIT-ML dataset (lower section). The symbol $^{\S}$ indicates that our text encoder is used to replace their original CLIP-text encoder.

text encoder focuses more on motion keywords, capturing multiple key movements and temporal information, yielding better results. Similarly, the importance of effective conditional encoders is being increasingly recognized in other domains, such as text-to-image generation and music-to-dance synthesis. In summary, simply replacing the original encoder with our method yields measurable performance improvements. This is an initial attempt. The motion-text field is rapidly evolving, with numerous related tasks emerging, such as Text to Human-Object Interaction(HOI) Motion Generation Xu et al. (2023), Text to Human-Scene Interaction (HSI) Motion Generation Cen et al. (2024), Text to Human-Human Interaction (HHI) Motion Generation Liang et al. (2024), and scenarios involving multiple people, controllable actions, long sequences, and variations in skeletal structures. These areas demonstrate significant potential. We aim to explore its application to a broader range of human motion-related tasks in the future.

## C.3 Complexity Analysis of the Model

**Computational Consumption of Model Training.** We compare the training time of different models on the HumanML3D dataset using a single A6000 GPU and report the results in Table 12. During the training process, TM2T, MotionGPT, and our proposed method all require pretraining the motion encoder, which involves higher computational costs. Notably, our method ($\tilde{4}0$ hours) is significantly faster than TM2T (4 days) and MotionGPT (over 4 days). This demonstrates that while our multi-phase training strategy introduces additional steps compared to simpler methods, it remains computationally efficient relative to many existing motion-language approaches. In contrast, methods like TMR and MotionPatch leverage a pre-trained motion encoder and only train the alignment phase, resulting in reduced computational demands. To ensure a fair comparison, we replaced the dense heads in these methods with our Lexical Disentanglement Head, denoting them as †TMR and †MotionPatch, respectively. Our evaluation shows that the computational costs of †TMR and †MotionPatch remain nearly identical to the original TMR and MotionPatch implementations. While we attempted to utilize a pre-trained motion encoder in our method to reduce computational demands, this configuration resulted in significantly degraded performance. Therefore, future work will focus on fine-tuning the pre-trained motion encoder with our Lexical Disentanglement Head. This approach has the potential to effectively balance computational efficiency and performance.

**Inference Efficiency.** It is also important to note that in practical text-to-motion retrieval scenarios, motion features are pre-extracted and stored, meaning retrieval efficiency primarily depends on the text encoder. As shown in Fig. 4, our method supports sparsification techniques that reduce feature dimensions, improving computational efficiency during retrieval. Additionally, as demon-

strated in Fig. 10, our model supports keyword-based searches by directly activating specific feature dimensions with weights set to 1, thereby eliminating the need for a text encoder during retrieval. This capability significantly improves efficiency for targeted queries. When combined with rapid search techniques such as inverted indexing, our approach offers substantial potential for optimizing retrieval speed and scalability.

| Methods | Training Epoch | Training Time |
|---------|----------------|---------------|
| T2M | 300 + 270 + 300 + 50 | ∼ 3 days |
| TMR | 500 | ∼ 20 minutes |
| MotionPatch | 50 | ∼ 30 minutes |
| † TMR | 500 | ∼ 20 minutes |
| † MotionPatch | 50 | ∼ 30 minutes |
| TM2T | 300 + 300 | ∼ 4 days |
| MotionGPT | 3000 | ≫ 4 days |
| Ours | 50 + 200 + 150 + 20 | ∼ 40 hours |

Table 12: Comparison of Training Epochs and Time on Motion-to-Text Captioning Benchmarks for HumanML3D. The symbol † indicates that the lexicon representation is directly used instead of a dense embedding. "≫ 4 days" signifies that the training time for MotionGPT greatly exceeds 4 days.

## C.4 ROBUSTNESS AND EFFECTIVENESS ANALYSIS OF THE LEARNED REPRESENTATION

We conducted two experiments on HumanML3D Guo et al. (2022a) and Motion-X Lin et al. (2024) to evaluate the robustness and effectiveness of the learned lexical representations. Since Motion-X includes data overlapping with HumanML3D, we removed the overlapping content to ensure a fair assessment of generalization.

**Small Dataset Training, Large Dataset Testing.** We trained our model on the smaller HumanML3D dataset and tested it on the larger Motion-X dataset. As shown in Table 14, our method outperforms existing approaches such as TMR and MotionPatch. Motion-X features more diverse scenarios, longer text descriptions, and richer semantics, which present challenges for methods like TMR and MotionPatch that rely on dense embeddings of global features. In contrast, our lexical representation captures keyword-level importance, allowing it to generalize better across diverse data.

**Large Dataset Training, Small Dataset Testing.** We trained our model on the larger Motion-X dataset and tested it on the smaller HumanML3D dataset. As shown in Table 13, all methods performed better than in the first experiment, underscoring the value of larger datasets. Importantly, our method continued to outperform others, further demonstrating its robustness and effectiveness.

However, unlike the image domain, where data is relatively easy to obtain, the human motion domain faces significant challenges in acquiring motion data. High-quality datasets primarily rely on motion capture (MoCap) systems. Datasets like KIT-ML and HumanML3D are among the largest and highest-quality MoCap datasets currently available. There are also datasets, such as Motion-X, that use algorithms to automatically extract human motion from videos. While this approach significantly expands dataset size, it often suffers from issues like jitter and anomalies. With future advancements in related technologies, it may become easier to extract more accurate human motion data from videos, enabling rapid dataset expansion. In addition, the quality of motion captions requires further improvement. As shown in Fig. 5(b), the distribution of keywords in datasets is imbalanced, and many descriptions are overly simplistic, failing to highlight meaningful differences. Phrases like "a person action" are common, resulting in significantly higher text similarity compared to other domains (e.g., as noted in TMR). While some datasets now include fine-grained annotations, many rely on AI-generated descriptions, such as those created by ChatGPT-like models, raising concerns about quality and consistency. Therefore, it is also necessary to deepen our understanding of human motion to improve dataset quality. Overall, constructing larger and higher-quality datasets remains a critical direction. We hope that our interpretable sparse lexical representations can enhance our understanding of motion, facilitating the creation of more accurate and detailed text annotations. This would improve the quality of datasets and advance human motion understanding and generation tasks.

| Methods | Text to motion retrieval | | | | | | Motion to text retrieval | | | | | |
|---------|------|------|------|------|-------|--------|------|------|------|------|-------|--------|
| | R@1↑ | R@2↑ | R@3↑ | R@5↑ | R@10↑ | MedR↓ | R@1↑ | R@2↑ | R@3↑ | R@5↑ | R@10↑ | MedR↓ |
| TMR | 0.06 | 0.11 | 0.24 | 0.39 | 0.51 | 829.00 | 0.08 | 0.16 | 0.21 | 0.37 | 0.56 | 861.50 |
| MotionPatch | 0.12 | 0.28 | 0.49 | 0.78 | 0.96 | 768.00 | 0.14 | 0.29 | 0.48 | 0.82 | 0.98 | 781.00 |
| Ours | 0.24 | 0.46 | 0.82 | 0.98 | 1.24 | 503.00 | 0.26 | 0.48 | 0.86 | 1.02 | 1.43 | 516.00 |

Table 13: Zero-Shot Text-to-Motion Retrieval Results. Methods are trained on HumanML3D and tested on Motion-X.

| Methods | Text to motion retrieval | | | | | | Motion to text retrieval | | | | | |
|---------|------|------|------|------|-------|--------|------|------|------|------|-------|--------|
| | R@1↑ | R@2↑ | R@3↑ | R@5↑ | R@10↑ | MedR↓ | R@1↑ | R@2↑ | R@3↑ | R@5↑ | R@10↑ | MedR↓ |
| TMR | 1.62 | 2.58 | 3.74 | 5.71 | 7.12 | 213.00 | 1.81 | 2.68 | 3.86 | 5.82 | 7.43 | 224.50 |
| MotionPatch | 1.86 | 2.78 | 3.92 | 5.87 | 7.96 | 201.00 | 1.92 | 2.84 | 4.12 | 6.03 | 7.64 | 208.50 |
| Ours | 2.76 | 3.82 | 5.13 | 7.42 | 8.76 | 182.00 | 2.83 | 3.96 | 5.28 | 7.89 | 8.98 | 186.00 |

Table 14: Zero-Shot Text-to-Motion Retrieval Results. Methods are trained on Motion-X and tested on HumanML3D.

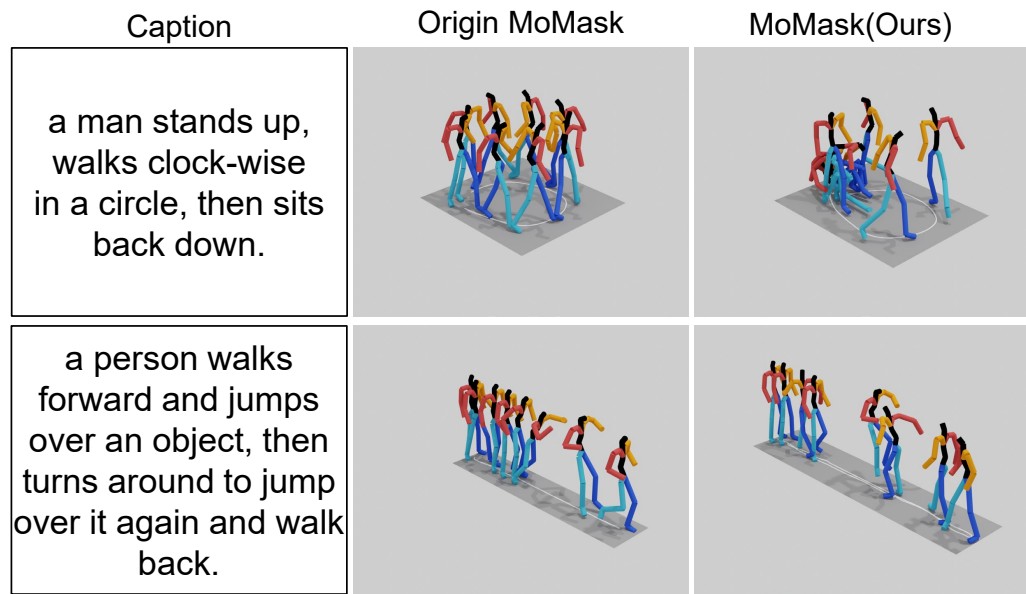

Figure 11: The Text to Motion Generation Results. Our text encoder replaces the original CLIP-text encoder in MoMask. When multiple actions are present in a sentence, the original MoMask often overlooks some actions, resulting in poor performance. This improvement arises because the pre-trained CLIP-text encoder does not prioritize motion-related information, and the lack of temporal information in images makes it less sensitive to the sequential nature of actions. In contrast, ours text encoder focuses more on motion keywords, capturing multiple key movements and temporal information, yielding better results.

# D    MORE DISCUSSIONS

From the perspective of socio-cognitive terminology, our research provides a new unit of understanding for human motion. From the viewpoint of the semantic triangle model and socio-cognitive terminology, we establish a strong connection between human motion and language. The semantic triangle model underscores the relationship between the mind, language, and the world, highlighting how symbols (language) mediate our understanding of the world and influence our cognitive processes. Socio-cognitive terminology further emphasizes how language reflects human cognition and social interaction. In our approach, language is not merely a tool for describing actions but serves as a key to linking natural language vocabulary with the features and patterns of motion. This enables language to act as a mediator, enriching our understanding and generation of human motion. Our work offers a fresh perspective on the interplay between motion and language, providing a solid foundation for future human motion understanding and generation.

## D.1    DOES THE PROPOSED ALIGNMENT PARADIGM GENERATE UNEVEN SEMANTIC DENSITY WITHIN THE JOINT FEATURE SPACE?

The proposed alignment paradigm does indeed reflect a certain degree of uneven semantic density within the joint feature space, primarily due to the inherent imbalance in the motion and language data.

From the perspective of socio-cognitive terminology, this uneven distribution can be seen as a result of the varying informational density across different lexical items, as seen in polysemy and metaphorical extensions. The lexicon in the dataset carries different levels of semantic richness, with words often being overloaded due to their metaphorical or contextual usage across a range of motion scenarios. For example, terms like "transition," which occur frequently, carry a higher semantic density, whereas more specific actions, such as "lick," appear less frequently and thus exhibit lower semantic density.

From an AI perspective, the semantic imbalance reflects the inherent challenges posed by the dataset's structure. Datasets like HumanML3D and KIT-ML utilize full-sentence annotations, which obscure the underlying imbalance in the semantic distribution. A clearer example can be found in the Babel Punnakkal et al. (2021) dataset, which, like HumanML3D, uses motion data from AMASS but annotates the motions using action classes rather than full sentences. Because Babel relies on action class annotations, it offers a more intuitive view of the distributional imbalances in the data. Babel's action classes exhibit a clear long-tailed distribution, with the most frequent action, "transition," occurring 17,287 times, while the 50th most frequent action, "sports moves," occurs 280 times, the 100th most frequent action, "communicate," occurs 52 times, and the 200th most frequent action, "lick," occurs only 5 times. The frequency of actions follows Zipf's law, underscoring the disparity in action occurrences across categories and highlighting the uneven semantic density within the dataset.

Our proposed alignment paradigm facilitates a shift towards a more interpretable and sparse lexical representation. This approach enables a clearer analysis of the relationship between different terms, as we can now observe how specific words and actions are distributed within the joint feature space. By transforming both motion and text into interpretable representations, we can better capture the semantic relationships between different actions and words. For instance, actions that often occur together, such as "eating food" and "raising right hand to mouth," share a closer semantic relationship than unrelated actions, such as "dancing" and "extending arms."

In conclusion, the proposed paradigm does reflect uneven semantic density in the joint feature space, but this is consistent with the underlying imbalance present in the data. The relationships between terms are influenced by their frequency and contextual usage, and further linguistic analysis could provide deeper insights into these connections, ultimately contributing to a better understanding and generation of human motion. Our method, by focusing on sparse lexical representations, allows us to examine and address these semantic disparities, paving the way for improved motion-text alignment and more effective human motion understanding and generation.

### D.2 What is the relation of different terms?

From the perspective of Socio-Cognitive Terminology, the relationships among terms in our lexicon can be understood through interconnected dimensions reflecting cognitive, linguistic, and cultural structures. Terms are related through synonymy and antonymy, where synonyms like "walking" and "strolling" capture similar concepts, while antonyms like "fast" and "slow" highlight oppositional dynamics in motion semantics. These relationships help associate related terms within shared semantic categories. Prototypicality and categorization further organize terms around central prototypes; for instance, "running" may serve as a prototype for rapid movements, with subcategories like "sprinting" and "jogging." Additionally, terms are grouped into broader categories, such as "physical actions" (e.g., "jump," "bend") and "social interactions" (e.g., "greet," "wave").

Terms also exhibit hierarchical relationships, where broader hypernyms like "movement" encompass specific hyponyms such as "running," "swimming," and "cycling." Cross-category links further connect terms from different domains, such as actions and emotions (e.g., "angry punching"). Cultural and contextual dependencies influence term meanings based on societal norms and specific scenarios, such as "handshake" varying across cultures or "intense activity" differing between sports and medical contexts. Moreover, these relationships are dynamic and adaptable, evolving with societal shifts and technological advancements; for example, new terms like "virtual gestures" may emerge in the metaverse.

By structuring terms around these relationships, our lexicon enhances motion understanding by capturing nuanced distinctions, facilitating semantic alignment between motion and text, and supporting tasks like motion captioning and retrieval. Ultimately, these interconnected relationships enable our model to align motion and text representations effectively, improving interpretability and robustness in motion-language tasks.

### D.3 How to understand the overlap between words and synonyms (verbs)?

The overlap between words and synonyms (e.g., verbs) in our lexical representation framework reflects the intrinsic flexibility and evolution of human language. From the perspective of Socio-Cognitive Terminology, this overlap serves both functional and interpretative roles, enabling our model to capture nuanced motion semantics and align text with motion effectively. Understanding this overlap requires examining the principles of polysemy and synonymy in the lexicon, as well as their implications for representation learning in the motion-language domain.

**The Functional Role of Polysemy and Synonymy.** Polysemy arises as a natural result of linguistic and conceptual evolution. Words gain additional meanings over time, often through mechanisms like metaphor and metonymy, creating layers of semantic variants clustered around a prototypical core. For instance, in the context of motion, terms like "walk" or "run" may develop nuanced meanings based on pace, style, or cultural interpretation. This flexibility enables a single lexical item to capture a range of related motions, enriching the representation's adaptability and interpretability. Similarly, synonymy reflects different perspectives on similar phenomena. Near-synonyms such as "jump," "leap," and "hop" provide slightly distinct viewpoints on upward motion, influenced by factors like intensity or purpose. In specialized discourse, this diversity allows for greater descriptive precision and contextual relevance. For example, "dancing waltz" may activate related terms like "spin," "circle," and "partner," each offering a complementary perspective on the action.

**Implications for Lexical Representation.** In our approach, the BERT vocabulary, constructed via Byte Pair Encoding (BPE), inherently includes synonyms and morphological variations. For instance, words like "walk," "walking," and "walked" co-exist within the lexicon, allowing the model to generalize across tenses and contexts. While this design effectively leverages existing linguistic diversity, we have not yet applied explicit optimizations to reduce redundancy or refine the representation of synonyms and polysemous terms. The overlap among terms is particularly evident during motion-text alignment. Lexical representations of motion-related queries often activate clusters of semantically related words. For example, a query involving "run" may also highlight terms like "sprint" or "dash," reflecting the shared semantic space in which motion concepts are embedded. This redundancy is not a flaw but rather a functional aspect that enhances the model's ability to handle diverse linguistic expressions of motion.

**Cognitive and Practical Perspectives.** From a cognitive semantics viewpoint, our representation system aligns with Geeraerts' theory of prototypical categorization. New lexical items and meanings are integrated into existing semantic structures without drastically altering the conceptual framework. For example, terms like "kick" or "strike" naturally connect to a core concept of forceful limb movement, while their specific contexts (e.g., "karate kick") expand the category's richness. Practically, this overlap aids in improving retrieval and captioning tasks. The lexical representation effectively captures nuances in motion semantics, outperforming traditional dense embeddings. The activation of synonyms and polysemous terms allows the model to retrieve motions or generate captions that align with diverse textual descriptions, ensuring robustness across tasks and datasets.

**Future Directions** While our current implementation demonstrates the utility of synonymy and polysemy, there is potential for further refinement. For instance, Socio-Cognitive Terminology research could inform new methods to optimize the BERT vocabulary for the motion-language domain, reducing redundancy while preserving semantic richness. Recent studies in natural language processing have explored vocabulary restructuring for improved model interpretability, and similar techniques could benefit motion-language tasks.

## D.4 SUMMARY

We sincerely appreciate the insightful comments of the reviewers, which have provided us with a fresh perspective by introducing the framework of Socio-Cognitive Terminology. From this vantage point, our research offers a new unit of understanding for human motion by aligning motion and text representations within a shared semantic space. This approach highlights the intricate interplay between linguistic structures and cognitive patterns, enabling a more nuanced interpretation of motion semantics. It is important to note that our work represents an initial exploration of this intersection and the need for further studies by experts in terminology science to address the unique challenges of defining and categorizing terms in the human motion domain. Just as Rita Temmerman's pioneering research in the life sciences laid the groundwork for the development of Socio-Cognitive Terminology, we hope our study serves as a stepping stone toward a deeper understanding of motion-related terminology. As Temmerman noted, understanding is a never-ending process. We aspire for our method to contribute meaningfully to the field, enhancing both the understanding and generation of human motion, and fostering further interdisciplinary research at the intersection of language, cognition, and motion.

