# OpenReview forum: "Towards Unified Human Motion-Language Understanding via Sparse Interpretable Characterization"
_ICLR.cc/2025/Conference — ICLR 2025 Poster_

### Official Review · Reviewer_DGmS · 2024-10-27

**Soundness:** 3
**Presentation:** 3
**Contribution:** 3
**Rating:** 8
**Confidence:** 4

**Summary:**

This paper presents a novel human motion-language pre-training framework that leverages lexical representation to enhance the interpretability of motion representations. The authors propose a multi-phase training strategy, including Lexical Bottlenecked Masked Language Modeling (LexMLM), Contrastive Masked Motion Modeling (CMMM), Lexical Bottlenecked Masked Motion Modeling (LexMMM), and Lexical Contrastive Motion-Language Pretraining (LexCMLP). The framework aims to align motion and text within a shared lexical vocabulary space, thereby improving the understanding of human motion. The authors demonstrate the effectiveness of their model through comprehensive experiments on multiple public datasets, showing state-of-the-art performance across various tasks and scenarios.

**Strengths:**

1. The paper is well-organized and easy to follow.
2. The paper introduces a pioneering approach to human motion-language understanding by creating a unified motion-language space that enhances interpretability through lexical representation. The proposed framework align both motion and text within a shared lexical vocabulary space, which can be trained by a multi-phase training strategy.
3. The authors have conducted extensive experiments and provided thorough analyses. The model achieves state-of-the-art results across multiple benchmarks, indicating the effectiveness of the proposed approach.
4. The proposed method enjoys better interpretability compared to existing method, which is further validated by the provided qualitative results.

**Weaknesses:**

1. Complexity of the Model: The multi-phase training strategy, while effective, may be overly complex and could potentially hinder reproducibility for researchers with limited resources. The authors could provide more details on the computational consumption for training the model compared to existing works.
2. One of the primary applications for motion-language pretraining is the text-to-motion generation task. What are the prospects of applying the proposed lexical-based method to this task?

**Questions:**

None

---

> ### Author Response · Authors · 2024-11-19
> **Response to DGmS (1/3)**
>
> _Thank you for your constructive and thoughtful comments. They were indeed helpful in improving the paper. We take this opportunity to address your concerns:_
>
> _**List of changes in the manuscript:**_
>
> > 1. `Section C.0.2 in Appendix` Extension to Text-to-Motion Generation.
> > 2. `Section C.0.3 in Appendix` Complexity Analysis of the Model
>
> ---
>
> > **W1: Complexity of the Model: The multi-phase training strategy, while effective, may be overly complex and could potentially hinder reproducibility for researchers with limited resources. The authors could provide more details on the computational consumption for training the model compared to existing works.**
>
> **A:**
>
> **Computational Consumption of Model Training.** We compare the training time of different models on the HumanML3D dataset using a single A6000 GPU and report the results in the following table.
>
> | Methods | Training Epoch | Training Time |
> | --- | --- | --- |
> | T2M | 300 + 270 + 300 + 50 | ~3 days |
> | TMR | 500 | ~20 minutes |
> | MotionPatch | 50 | ~30 minutes |
> | $ \dagger $ TMR | 500 | ~20 minutes |
> | $ \dagger $ MotionPatch | 50 | ~30 minutes |
> | TM2T | 300 + 300 | ~4 days |
> | MotionGPT | 3000 | $ \gg $ 4 days |
> | Ours | 50 + 200 + 150 + 20 | ~ 40 hours |
>
>
> During the training process, TM2T, MotionGPT, and our proposed method all require pretraining the motion encoder, which involves higher computational costs. Notably, our method (~40 hours) is significantly faster than TM2T (4 days) and MotionGPT (over 4 days). This demonstrates that while our multi-phase training strategy introduces additional steps compared to simpler methods, it remains computationally efficient relative to many existing motion-language approaches.
>
> In contrast, methods like TMR and MotionPatch leverage a pre-trained motion encoder and only train the alignment phase, resulting in reduced computational demands. To ensure a fair comparison, we replaced the dense heads in these methods with our Lexical Disentanglement Head, denoting them as $ \dagger $TMR and $ \dagger $MotionPatch, respectively. Our evaluation shows that the computational costs of $ \dagger $TMR and $ \dagger $MotionPatch remain nearly identical to the original TMR and MotionPatch implementations.
>
> While we attempted to utilize a pre-trained motion encoder in our method to reduce computational demands, this configuration resulted in significantly degraded performance. Therefore, future work will focus on fine-tuning the pre-trained motion encoder with our Lexical Disentanglement Head. This approach has the potential to effectively balance computational efficiency and performance.
>
> **Inference Efficiency.** It is also important to note that in practical text-to-motion retrieval scenarios, motion features are pre-extracted and stored, meaning retrieval efficiency primarily depends on the text encoder. As shown in Figure 4, our method supports sparsification techniques that reduce feature dimensions, improving computational efficiency during retrieval.
>
> Additionally, as demonstrated in Figure 10, our model supports keyword-based searches by directly activating specific feature dimensions with weights set to 1, thereby eliminating the need for a text encoder during retrieval. This capability significantly improves efficiency for targeted queries. When combined with rapid search techniques such as inverted indexing, our approach offers substantial potential for optimizing retrieval speed and scalability.

---

> ### Author Response · Authors · 2024-11-19
> **Response to DGmS (2/3)**
>
> > **W2: One of the primary applications for motion-language pretraining is the text-to-motion generation task. What are the prospects of applying the proposed lexical-based method to this task?**
>
> **A:** We sincerely thank the reviewer for their thoughtful suggestion to strengthen our paper.  Our method can be seamlessly integrated into text-to-motion (T2M) generation tasks. Specifically, we directly replace the text encoder in existing methods with our proposed lexical text encoder to extract a more refined and sparse lexical representation. We observe consistent performance improvements across various generative architectures by integrating our method.
>
> Comparative experiments were conducted on multiple frameworks, including VAE-based methods (e.g., T2M`[3]`), diffusion-based approaches (e.g., MDM`[4]`), autoregressive models (e.g., T2M-GPT`[5]`),  and non-autoregressive models (e.g., MoMask`[6]`).
>
> The results, summarized in the following tables, show notable gains, particularly in metrics evaluating motion-text alignment, such as R-precision (Top-1, Top-2, Top-3 accuracy) and Multimodal Distance (MM-Dist). These improvements are primarily attributed to two factors: (1) CLIP-based text encoders, pre-trained on image-text datasets, do not sufficiently focus on motion-related content, and (2) these encoders fail to capture temporal information during the text-image pretraining process. In contrast, our text encoder is specifically designed to identify motion-relevant keywords during pretraining and align them with a motion encoder that captures spatiotemporal motion information, leading to enhanced performance in T2M tasks. This aligns with recent observations in specialized T2M studies, such as HumanTomato`[1]` and LGTM`[2]`, which highlight similar limitations of original CLIP text encoders. However, the majority of mainstream T2M research still emphasizes improving generation paradigms to achieve better results.
>
> | **Dataset** | **Paradigms** | **Methods** | **FID ↓** | **Top1 ↑** | **Top2 ↑** | **Top3 ↑** | **MM-Dist ↓** |
> | --- | --- | --- | --- | --- | --- | --- | --- |
> | **HumanML3D** | **VAE** | T2M`[3]` | $ 1.087^{\pm .021} $ | $ 0.455^{\pm .003} $ | $ 0.636^{\pm .003} $ | $ 0.736^{\pm .002} $ | $ 3.347^{\pm .008} $ |
> |  |  | **T2M**$ ^{\S} $ | $ 0.942^{\pm .009} $ | $ 0.472^{\pm .004} $ | $ 0.653^{\pm .002} $ | $ 0.748^{\pm .003} $ | $ 3.104^{\pm .006} $ |
> |  | **Diffusion** | MDM`[4]` | $ 0.544^{\pm .044} $ | $ 0.320^{\pm .005} $ | $ 0.498^{\pm .004} $ | $ 0.611^{\pm .007} $ | $ 5.566^{\pm .027} $ |
> |  |  | **MDM**$ ^{\S} $ | $ 0.524^{\pm .036} $ | $ 0.357^{\pm .004} $ | $ 0.536^{\pm .003} $ | $ 0.643^{\pm .005} $ | $ 5.212^{\pm .021} $ |
> |  | **AR** | T2M-GPT`[5]` | $ 0.141^{\pm .005} $ | $ 0.492^{\pm .003} $ | $ 0.679^{\pm .002} $ | $ 0.775^{\pm .002} $ | $ 3.121^{\pm .009} $ |
> |  |  | **T2M-GPT**$ ^{\S} $ | $ 0.133^{\pm .005} $ | $ 0.506^{\pm .004} $ | $ 0.684^{\pm .003} $ | $ 0.781^{\pm .004} $ | $ 3.002^{\pm .006} $ |
> |  | **NAR** | MoMask`[6]` | $ 0.045^{\pm .002} $ | $ 0.521^{\pm .002} $ | $ 0.713^{\pm .002} $ | $ 0.807^{\pm .002} $ | $ 2.958^{\pm .008} $ |
> |  |  | **MoMask**$ ^{\S} $ | $ \mathbf{0.041}^{\pm .002} $ | $ \mathbf{0.532}^{\pm .002} $ | $ \mathbf{0.721}^{\pm .003} $ | $ \mathbf{0.814}^{\pm .002} $ | $ \mathbf{2.852}^{\pm .008} $ |
>
>
> | **Dataset** | **Paradigms** | **Methods** | **FID ↓** | **Top1 ↑** | **Top2 ↑** | **Top3 ↑** | **MM-Dist ↓** |
> | --- | --- | --- | --- | --- | --- | --- | --- |
> | **KIT-ML** | **VAE** | T2M`[3]` | $ 3.022^{\pm .107} $ | $ 0.361^{\pm .005} $ | $ 0.559^{\pm .007} $ | $ 0.681^{\pm .007} $ | $ 3.488^{\pm .028} $ |
> |  |  | **T2M**$ ^{\S} $ | $ 2.836^{\pm .062} $ | $ 0.372^{\pm .004} $ | $ 0.574^{\pm .004} $ | $ 0.695^{\pm .005} $ | $ 3.235^{\pm .016} $ |
> |  | **Diffusion** | MDM`[4]` | $ 0.497^{\pm .021} $ | $ 0.164^{\pm .004} $ | $ 0.291^{\pm .004} $ | $ 0.396^{\pm .004} $ | $ 9.191^{\pm .022} $ |
> |  |  | **MDM**$ ^{\S} $ | $ 0.482^{\pm .009} $ | $ 0.214^{\pm .005} $ | $ 0.319^{\pm .004} $ | $ 0.418^{\pm .005} $ | $ 8.682^{\pm .014} $ |
> |  | **AR** | T2M-GPT`[5]` | $ 0.514^{\pm .029} $ | $ 0.416^{\pm .006} $ | $ 0.627^{\pm .006} $ | $ 0.745^{\pm .006} $ | $ 3.007^{\pm .023} $ |
> |  |  | **T2M-GPT**$ ^{\S} $ | $ 0.502^{\pm .016} $ | $ 0.423^{\pm .005} $ | $ 0.641^{\pm .006} $ | $ 0.752^{\pm .006} $ | $ 2.927^{\pm .015} $ |
> |  | **NAR** | MoMask`[6]` | $ 0.204^{\pm .011} $ | $ 0.433^{\pm .007} $ | $ 0.656^{\pm .005} $ | $ 0.781^{\pm .005} $ | $ 2.779^{\pm .022} $ |
> |  |  | **MoMask**$ ^{\S} $ | $ \mathbf{0.186}^{\pm .009} $ | $ \mathbf{0.441}^{\pm .006} $ | $ \mathbf{0.668}^{\pm .004} $ | $ \mathbf{0.792}^{\pm .005} $ | $ \mathbf{2.693}^{\pm .013} $ |

---

> ### Author Response · Authors · 2024-11-19
> **Response to DGmS (3/3)**
>
> Similarly, the significance of effective conditional encoders has been increasingly acknowledged in other domains, such as text-to-image generation and music-to-dance synthesis. Exploring advanced text encoders and representation learning techniques in the motion-text domain appears to be a highly promising direction.
>
> In summary, simply replacing the original encoder with our method yields measurable performance improvements. This is an initial attempt. The motion-text field is rapidly evolving, with numerous related tasks emerging, such as Text to Human-Object Interaction(HOI) Motion Generation`[7]`, Text to Human-Scene Interaction (HSI) Motion Generation`[8]`,  Text to Human-Human Interaction (HHI) Motion Generation`[9]`, and scenarios involving multiple people, controllable actions, long sequences, and variations in skeletal structures. These areas demonstrate significant potential. We aim to explore its application to a broader range of human motion-related tasks in the future.
>
> ----
>
> **Reference**
>
> `[1]`Lu S, Chen L H, Zeng A, et al. HumanTOMATO: Text-aligned Whole-body Motion Generation[C]//Forty-first International Conference on Machine Learning.
>
> `[2]`Sun H, Zheng R, Huang H, et al. LGTM: Local-to-Global Text-Driven Human Motion Diffusion Model[C]//ACM SIGGRAPH 2024 Conference Papers. 2024: 1-9.
>
> `[3]`Guo C, Zou S, Zuo X, et al. Generating diverse and natural 3d human motions from text[C]//Proceedings of the IEEE/CVF Conference on Computer Vision and Pattern Recognition. 2022: 5152-5161.
>
> `[4]`Guy Tevet, Sigal Raab, Brian Gordon, Yonatan Shafir, Daniel Cohen-Or, and Amit Haim Bermano. Human motion diffusion model. In Proceedings of the International Conference on Learning Representations. OpenReview.net, 2023.
>
> `[5]`Zhang J, Zhang Y, Cun X, et al. Generating human motion from textual descriptions with discrete representations[C]//Proceedings of the IEEE/CVF conference on computer vision and pattern recognition. 2023: 14730-14740.
>
> `[6]`Guo C, Mu Y, Javed M G, et al. Momask: Generative masked modeling of 3d human motions[C]//Proceedings of the IEEE/CVF Conference on Computer Vision and Pattern Recognition. 2024: 1900-1910.
>
> `[7]`Xu S, Li Z, Wang Y X, et al. Interdiff: Generating 3d human-object interactions with physics-informed diffusion[C]//Proceedings of the IEEE/CVF International Conference on Computer Vision. 2023: 14928-14940.
>
> `[8]`Cen Z, Pi H, Peng S, et al. Generating Human Motion in 3D Scenes from Text Descriptions[C]//Proceedings of the IEEE/CVF Conference on Computer Vision and Pattern Recognition. 2024: 1855-1866.
>
> `[9]`Liang H, Zhang W, Li W, et al. Intergen: Diffusion-based multi-human motion generation under complex interactions[J]. International Journal of Computer Vision, 2024: 1-21.

---

> > ### Comment · Reviewer_DGmS · 2024-11-20
> >
> > Thanks for the response. The authors have addressed my concerns and I'll maintain my rating.

---

> ### Author Response · Authors · 2024-11-20
>
> We sincerely appreciate your time and constructive comments throughout this process. Thank you for your positive feedback on the strengths of our work, including its clear organization, methodological novelty, strong performance, good presentation, and clear interpretability and visualization. Your recognition of these contributions motivates us to explore broader applications for our framework. Thank you again.

---

### Official Review · Reviewer_jkCN · 2024-11-01

**Soundness:** 3
**Presentation:** 2
**Contribution:** 3
**Rating:** 6
**Confidence:** 4

**Summary:**

This paper proposed a new method to better align text and motion via lexical representation contrastive learning. To address the problem of semantic deficiency, rough alignment between dense text and motion, etc., the authors proposed several new methods, i.e., LexMLM, CMMM, LexMMM, and LexCMLP to build a new pipeline. On several tasks and benchmarks, the proposed method performed well and verified the effectiveness of the proposed alignment algorithm.

**Strengths:**

+ The motivation of this paper is sound and non-trivial, the alignment of motion and text is vital for better joint representation learning of human motion and language.

+ The adopted methods make sense and show decent performance on widely used benchmarks and tasks.

+ Fig 5 shows well visualization.

**Weaknesses:**

- Need more discussions to better clarify the method, e.g., does the proposed alignment paradigm generate uneven semantic density within the joint feature space? What is the relation of different terms? How to understand the overlap between words and synonyms (verbs)?

- Some presentation details:

-- Fig 1: please add the variables of features in the alignment part.

-- What is the detailed implementation of the visualization of lexical representation?

-- Fig 2: hard to follow, please add more main text-figure corresponses, to help the readers to understand the detailed methodology.

- Fig 3: fonts are too small.

- Fig 6: hard to distinguish the sequential motions, there is too much overlap. Please use more frames or a larger interval.

- Fig 7: fonts are too small.

- Fig 8: please add more captions to clarify the comparison and details, e.g., what do the red fonts mean?

**Questions:**

Some typos:

- L148: pre-trained language models**space**(PLMs)

- L192: f a language model head**space**(LM-Head)

---

> ### Author Response · Authors · 2024-11-19
> **Response to jkCN (1/5)**
>
> _Thank you for our constructive and thoughtful comments. They were indeed helpful in improving the paper. We take this opportunity to address your concerns:_
>
> _**List of changes in the manuscript:**_
>
> > 1. `Figure 1` We redrew the figure as requested and aligned the relevant content.
> > 2. `Figure 2` We provide a more detailed caption.
> > 3. `Figure 3` We have increased the font size.
> > 4. `Figure 6` We have fixed it.
> > 5. `Figure 7` We have increased the font size.
> > 6. `Figure 9` We have updated the caption. The red fonts mean the keywords.
> > 7. `Line 148` We have fixed the typo.
> > 8. `Line 192` We have fixed the typo.
> >
>
> ---
>
> > **W1: Need more discussions to better clarify the method, e.g., does the proposed alignment paradigm generate uneven semantic density within the joint feature space? What is the relation of different terms? How to understand the overlap between words and synonyms (verbs)?**
> >
>
> **A:** Thank you for your insightful questions that advance our understanding of human motion in the context of language and terminology. We have spent a significant amount of time reading classical works in **socio-cognitive terminology**to better understand and answer these comments. In this response, we attempt to bridge artificial intelligence and cognitive **socio-cognitive terminology**, offering an exploratory answer in the hope that it provides some helpful insights.
>
> **From the perspective of socio-cognitive terminology, our research provides a new unit of understanding for human motion.**  From the viewpoint of the semantic triangle model and socio-cognitive terminology, we establish a strong connection between human motion and language. The semantic triangle model underscores the relationship between the mind, language, and the world, highlighting how symbols (language) mediate our understanding of the world and influence our cognitive processes. Socio-cognitive terminology further emphasizes how language reflects human cognition and social interaction. In our approach, language is not merely a tool for describing actions but serves as a key to linking natural language vocabulary with the features and patterns of motion. This enables language to act as a mediator, enriching our understanding and generation of human motion. Our work offers a fresh perspective on the interplay between motion and language, providing a solid theoretical foundation for future human motion understanding and generation.

---

> ### Author Response · Authors · 2024-11-19
> **Response to jkCN (2/5)**
>
> _**(1/3) Does the proposed alignment paradigm generate uneven semantic density within the joint feature space?**_
>
> The proposed alignment paradigm does indeed reflect a certain degree of uneven semantic density within the joint feature space, primarily due to the inherent imbalance in the motion and language data.
>
> From the perspective of socio-cognitive terminology, this uneven distribution can be seen as a result of the varying informational density across different lexical items, as seen in polysemy and metaphorical extensions. The lexicon in the dataset carries different levels of semantic richness, with words often being overloaded due to their metaphorical or contextual usage across a range of motion scenarios. For example, terms like "transition," which occur frequently, carry a higher semantic density, whereas more specific actions, such as "lick," appear less frequently and thus exhibit lower semantic density.
>
> From an AI perspective, the semantic imbalance reflects the inherent challenges posed by the dataset's structure. Datasets like HumanML3D and KIT-ML utilize full-sentence annotations, which obscure the underlying imbalance in the semantic distribution. A clearer example can be found in the Babel`[1]`  dataset, which, like HumanML3D, uses motion data from AMASS but annotates the motions using action classes rather than full sentences. Because Babel relies on action class annotations, it offers a more intuitive view of the distributional imbalances in the data.  Babel’s action classes exhibit a clear long-tailed distribution, with the most frequent action, "transition," occurring 17,287 times, while the 50th most frequent action, "sports moves," occurs 280 times, the 100th most frequent action, "communicate," occurs 52 times, and the 200th most frequent action, "lick," occurs only 5 times.   The frequency of actions follows Zipf’s law, underscoring the disparity in action occurrences across categories and highlighting the uneven semantic density within the dataset.
>
> Our proposed alignment paradigm facilitates a shift towards a more interpretable and sparse lexical representation. This approach enables a clearer analysis of the relationship between different terms, as we can now observe how specific words and actions are distributed within the joint feature space. By transforming both motion and text into interpretable representations, we can better capture the semantic relationships between different actions and words. For instance, actions that often occur together, such as "eating food" and "raising right hand to mouth," share a closer semantic relationship than unrelated actions, such as "dancing" and "extending arms."
>
> In conclusion, the proposed paradigm does reflect uneven semantic density in the joint feature space, but this is consistent with the underlying imbalance present in the data. The relationships between terms are influenced by their frequency and contextual usage, and further linguistic analysis could provide deeper insights into these connections, ultimately contributing to a better understanding and generation of human motion. Our method, by focusing on sparse lexical representations, allows us to examine and address these semantic disparities, paving the way for improved motion-text alignment and more effective human motion understanding and generation.

---

> ### Author Response · Authors · 2024-11-19
> **Response to jkCN (3/5)**
>
> _**(2/3) What is the relation of different terms?**_
>
> From the perspective of Socio-Cognitive Terminology, the relationships among terms in our lexicon can be understood through interconnected dimensions reflecting cognitive, linguistic, and cultural structures. Terms are related through **synonymy and antonymy**, where synonyms like "walking" and "strolling" capture similar concepts, while antonyms like "fast" and "slow" highlight oppositional dynamics in motion semantics. These relationships help associate related terms within shared semantic categories. **Prototypicality and categorization** further organize terms around central prototypes; for instance, "running" may serve as a prototype for rapid movements, with subcategories like "sprinting" and "jogging." Additionally, terms are grouped into broader categories, such as "physical actions" (e.g., "jump," "bend") and "social interactions" (e.g., "greet," "wave").
>
>
>
> Terms also exhibit **hierarchical relationships**, where broader hypernyms like "movement" encompass specific hyponyms such as "running," "swimming," and "cycling." Cross-category links further connect terms from different domains, such as actions and emotions (e.g., "angry punching"). **Cultural and contextual dependencies** influence term meanings based on societal norms and specific scenarios, such as "handshake" varying across cultures or "intense activity" differing between sports and medical contexts. Moreover, these relationships are **dynamic and adaptable**, evolving with societal shifts and technological advancements; for example, new terms like "virtual gestures" may emerge in the metaverse.
>
>
>
> By structuring terms around these relationships, our lexicon enhances motion understanding by capturing nuanced distinctions, facilitating semantic alignment between motion and text, and supporting tasks like motion captioning and retrieval. Ultimately, these interconnected relationships enable our model to align motion and text representations effectively, improving interpretability and robustness in motion-language tasks.

---

> ### Author Response · Authors · 2024-11-19
> **Response to jkCN (4/5)**
>
> _**(3/3) How to understand the overlap between words and synonyms (verbs)?**_
>
> The overlap between words and synonyms (e.g., verbs) in our lexical representation framework reflects the intrinsic flexibility and evolution of human language. From the perspective of Socio-Cognitive Terminology, this overlap serves both functional and interpretative roles, enabling our model to capture nuanced motion semantics and align text with motion effectively. Understanding this overlap requires examining the principles of polysemy and synonymy in the lexicon, as well as their implications for representation learning in the motion-language domain.
>
> **(a) The Functional Role of Polysemy and Synonymy**
>
> Polysemy arises as a natural result of linguistic and conceptual evolution. Words gain additional meanings over time, often through mechanisms like metaphor and metonymy, creating layers of semantic variants clustered around a prototypical core. For instance, in the context of motion, terms like "walk" or "run" may develop nuanced meanings based on pace, style, or cultural interpretation. This flexibility enables a single lexical item to capture a range of related motions, enriching the representation's adaptability and interpretability.
>
> Similarly, synonymy reflects different perspectives on similar phenomena. Near-synonyms such as "jump," "leap," and "hop" provide slightly distinct viewpoints on upward motion, influenced by factors like intensity or purpose. In specialized discourse, this diversity allows for greater descriptive precision and contextual relevance. For example, "dancing waltz" may activate related terms like "spin," "circle," and "partner," each offering a complementary perspective on the action.
>
> **(b) Implications for Lexical Representation**
>
> In our approach, the BERT vocabulary, constructed via Byte Pair Encoding (BPE), inherently includes synonyms and morphological variations. For instance, words like "walk," "walking," and "walked" co-exist within the lexicon, allowing the model to generalize across tenses and contexts. While this design effectively leverages existing linguistic diversity, we have not yet applied explicit optimizations to reduce redundancy or refine the representation of synonyms and polysemous terms.
>
> The overlap among terms is particularly evident during motion-text alignment.  Lexical representations of motion-related queries often activate clusters of semantically related words. For example, a query involving "run" may also highlight terms like "sprint" or "dash," reflecting the shared semantic space in which motion concepts are embedded. This redundancy is not a flaw but rather a functional aspect that enhances the model’s ability to handle diverse linguistic expressions of motion.
>
> **(c) Cognitive and Practical Perspectives**
>
> From a cognitive semantics viewpoint, our representation system aligns with Geeraerts' theory of prototypical categorization. New lexical items and meanings are integrated into existing semantic structures without drastically altering the conceptual framework. For example, terms like "kick" or "strike" naturally connect to a core concept of forceful limb movement, while their specific contexts (e.g., "karate kick") expand the category's richness.
>
> Practically, this overlap aids in improving retrieval and captioning tasks.  The lexical representation effectively captures nuances in motion semantics, outperforming traditional dense embeddings. The activation of synonyms and polysemous terms allows the model to retrieve motions or generate captions that align with diverse textual descriptions, ensuring robustness across tasks and datasets.
>
> **(d) Future Directions**
>
> While our current implementation demonstrates the utility of synonymy and polysemy, there is potential for further refinement. For instance, Socio-Cognitive Terminology research could inform new methods to optimize the BERT vocabulary for the motion-language domain, reducing redundancy while preserving semantic richness. Recent studies in natural language processing have explored vocabulary restructuring for improved model interpretability, and similar techniques could benefit motion-language tasks.

---

> ### Author Response · Authors · 2024-11-19
> **Response to jkCN (5/5)**
>
> _**Summary of the Response to W1**_
>
> We sincerely appreciate this insightful comment, which has provided us with a fresh perspective by introducing the framework of Socio-Cognitive Terminology. From this vantage point, our research offers a new unit of understanding for human motion by aligning motion and text representations within a shared semantic space. This approach highlights the intricate interplay between linguistic structures and cognitive patterns, enabling a more nuanced interpretation of motion semantics.  It is important to note that our work represents an initial exploration of this intersection and the need for further studies by experts in terminology science to address the unique challenges of defining and categorizing terms in the human motion domain. Just as Rita Temmerman's pioneering research in the life sciences laid the groundwork for the development of Socio-Cognitive Terminology, we hope our study serves as a stepping stone toward a deeper understanding of motion-related terminology.   As Temmerman noted, understanding is a never-ending process. We aspire for our method to contribute meaningfully to the field, enhancing both the understanding and generation of human motion, and fostering further interdisciplinary research at the intersection of language, cognition, and motion.
>
> ----
>
> > **W2: Some presentation details.**
>
> **A:** The word cloud is a visualization technique used to highlight the most frequently occurring words in text data. It displays words in varying sizes and colors, making key information easily noticeable. In our approach, we extract the lexical representations of motion and text, where each dimension corresponds to the importance of a particular word. We then use a word cloud to visualize these representations. The Python code is as follows.
>
> ```python
> import matplotlib.pyplot as plt
> from wordcloud import WordCloud
>
> word_importance = xxxx # {walk: 0.4, run:0.2}
> wordcloud = WordCloud(width=xxx, height=xxxx).generate_from_frequencies(word_importance)
> plt.imshow(wordcloud, interpolation='bilinear')
> plt.axis('off')
> plt.savefig('xxx_wordcloud.png', bbox_inches='tight', pad_inches=0, dpi=150)
> plt.close()
> ```
>
>
>
> > **W3: Modification of some figures.**
>
> **A:**
>
> > **Fig 1: please add the variables of features in the alignment part.**
>
> We redrew the figure as requested and aligned the relevant content.
>
> > **Fig 2: hard to follow, please add more main text-figure corresponses, to help the readers to understand the detailed methodology.**
>
> The framework of our method, including i) LexMLM, which enhances the language model's focus on high-entropy motion-related words; ii) CMMM, which captures spatial and temporal dynamics for robust motion representation; iii) LexMMM, which enables the motion model to identify semantic features and improve cross-modal understanding; and iv) LexCMLP, which aligns motion and text within a unified vocabulary space, ensuring cross-modal coherence。
>
> > **Fig 3: fonts are too small.**
>
> We have increased the font size.
>
> > **Fig 6: hard to distinguish the sequential motions, there is too much overlap. Please use more frames or a larger interval.**
>
> We have fixed it.
>
> > **Fig 7: fonts are too small.**
>
> We have increased the font size.
>
> > **Fig 8: please add more captions to clarify the comparison and details, e.g., what do the red fonts mean?**
>
>  We have fixed it. The red fonts mean the keywords.
>
> ----
>
> > **Q1: Some typos.**
>
> **A:** We have corrected these typos.
>
> > **L148: pre-trained language modelsspace(PLMs)**
>
> pre-trained language models (PLMs) space
>
> > **L192: f a language model headspace(LM-Head)**
>
>  a language model head (LM-Head) space
>
> ----
>
> **Reference**
>
> `[1]`Punnakkal A R, Chandrasekaran A, Athanasiou N, et al. BABEL: Bodies, action and behavior with english labels[C]//Proceedings of the IEEE/CVF Conference on Computer Vision and Pattern Recognition. 2021: 722-731.

---

> > ### Comment · Reviewer_jkCN · 2024-11-20
> > **Post-rebuttal**
> >
> > Thanks for the response. My concerns are mainly addressed. Please add the above additional discussion in the final version. I stand by my rating.

---

> > > ### Author Response · Authors · 2024-11-20
> > >
> > > We sincerely appreciate this insightful comment, which offers a fresh perspective on our work. We will incorporate the above discussion into the Appendix in the final version of the paper. Thank you once again for your valuable feedback.

---

### Official Review · Reviewer_uh5q · 2024-11-04

**Soundness:** 3
**Presentation:** 3
**Contribution:** 3
**Rating:** 6
**Confidence:** 3

**Summary:**

This paper introduces the lexical representation paradigm to the motion and language domain, where both the motion and language are mapped into a shared vocabulary space to enhance the interpretability of motion representations. A novel multi-phase pre-training framework is proposed to learned aligned, semantically correct sparse lexicon representation for both language and motion modalities. Comprehensive experiments show that the model achieves state-of-the-art performance across various tasks and scenarios.

**Strengths:**

1. Introducing the lexical representation paradigm to the text-motion domain is novel;
2. Experimental results show remarkable improvements over baseline methods;
3. The paper is well written and easy to follow.

**Weaknesses:**

1. The lexical representation of motion and text is trained on small datasets, such as HumanML3D and KIT-ML. Are these datasets enough to learn good lexical representations? Generalization experiments are desired to show the robustness and effectiveness of the learned representation.
2. Interpretability of motion representation: From the visualization results in Figure 5, it seems that there exist some meaningless words in the word cloud visualization result. Does that mean that the representation is not compact or noisy?
3. In Figure 2, the caption of the four modules is out of order.

**Questions:**

Please refer to the weaknesses for my main concerns.

---

> ### Author Response · Authors · 2024-11-19
> **Response to uh5q (1/2)**
>
> _**List of changes in the manuscript:**_
>
> > 1. `Figure 2` We have fixed the order.
> > 2. `Section C.0.4 in Appendix` Robustness and Effectiveness of the Learned Representation.
>
> ----
>
> > **W1: The lexical representation of motion and text is trained on small datasets, such as HumanML3D and KIT-ML. Are these datasets enough to learn good lexical representations? Generalization experiments are desired to show the robustness and effectiveness of the learned representation.**
> >
>
> **A:** We conducted experiments on HumanML3D and Motion-X to evaluate the robustness and effectiveness of the learned lexical representations. Since Motion-X includes data overlapping with HumanML3D, we removed the overlapping content to ensure a fair assessment of generalization.
>
> 1. **Small Dataset Training, Large Dataset Testing:**
> We trained our model on the smaller HumanML3D dataset and tested it on the larger Motion-X dataset. As shown in the following table, our method outperforms existing approaches such as TMR and MotionPatch. Motion-X features more diverse scenarios, longer text descriptions, and richer semantics, which present challenges for methods like TMR and MotionPatch that rely on dense embeddings of global features. In contrast, our lexical representation captures keyword-level importance, allowing it to generalize better across diverse data.
>
> | Methods |  |  Text to | Motion |  Retrieval |      | |  | Motion | to Text | Retrieval | |   |
> | --- | --- | --- | --- | --- | --- | --- | --- | --- | --- | --- | --- | --- |
> |  | R@1 $ \uparrow $ | R@2 $ \uparrow $ | R@3 $ \uparrow $ | R@5 $ \uparrow $ | R@10 $ \uparrow $ | MedR $ \downarrow $ | R@1 $ \uparrow $ | R@2 $ \uparrow $ | R@3 $ \uparrow $ | R@5 $ \uparrow $ | R@10 $ \uparrow $ | MedR $ \downarrow $ |
> | TMR | 0.06 | 0.11 | 0.24 | 0.39 | 0.51 | 829.00 | 0.08 | 0.16 | 0.21 | 0.37 | 0.56 | 861.50 |
> | MotionPatch | 0.12 | 0.28 | 0.49 | 0.78 | 0.96 | 768.00 | 0.14 | 0.29 | 0.48 | 0.82 | 0.98 | 781.00 |
> | Ours | 0.24 | 0.46 | 0.82 | 0.98 | 1.24 | 503.00 | 0.26 | 0.48 | 0.86 | 1.02 | 1.43 | 516.00 |
>
>
> 2. **Large Dataset Training, Small Dataset Testing:**
> We trained our model on the larger Motion-X dataset and tested it on the smaller HumanML3D dataset. As shown in the following table, all methods performed better than in the first experiment, underscoring the value of larger datasets. Importantly, our method continued to outperform others, further demonstrating its robustness and effectiveness.
>
> | Methods |   | Text to  | Motion  | Retrieval | |   |   | Motion  | to Text  | Retrieval |   |  |
> | --- | --- | --- | --- | --- | --- | --- | --- | --- | --- | --- | --- | --- |
> |  | R@1 $ \uparrow $ | R@2 $ \uparrow $ | R@3 $ \uparrow $ | R@5 $ \uparrow $ | R@10 $ \uparrow $ | MedR $ \downarrow $ | R@1 $ \uparrow $ | R@2 $ \uparrow $ | R@3 $ \uparrow $ | R@5 $ \uparrow $ | R@10 $ \uparrow $ | MedR $ \downarrow $ |
> | TMR | 1.62 | 2.58 | 3.74 | 5.71 | 7.12 | 213.00 | 1.81 | 2.68 | 3.86 | 5.82 | 7.43 | 224.50 |
> | MotionPatch | 1.86 | 2.78 | 3.92 | 5.87 | 7.96 | 201.00 | 1.92 | 2.84 | 4.12 | 6.03 | 7.64 | 208.50 |
> | Ours | 2.76 | 3.82 | 5.13 | 7.42 | 8.76 | 182.00 | 2.83 | 3.96 | 5.28 | 7.89 | 8.98 | 186.00 |
>
> However, unlike the image domain, where data is relatively easy to obtain, the human motion domain faces significant challenges in acquiring motion data. High-quality datasets primarily rely on motion capture (MoCap) systems. Datasets like KIT-ML and HumanML3D are among the largest and highest-quality MoCap datasets currently available. There are also datasets, such as Motion-X, that use algorithms to automatically extract human motion from videos. While this approach significantly expands dataset size, it often suffers from issues like jitter and anomalies. With future advancements in related technologies, it may become easier to extract more accurate human motion data from videos, enabling rapid dataset expansion.
>
> In addition, the quality of motion captions requires further improvement. As shown in Fig.5(b), the distribution of keywords in datasets is imbalanced, and many descriptions are overly simplistic, failing to highlight meaningful differences. Phrases like "a person action" are common, resulting in significantly higher text similarity compared to other domains (e.g., as noted in TMR). While some datasets now include fine-grained annotations, many rely on AI-generated descriptions, such as those created by ChatGPT-like models, raising concerns about quality and consistency.  Therefore, it is also necessary to deepen our understanding of human motion to improve dataset quality.
>
> Overall, constructing larger and higher-quality datasets remains a critical direction. We hope that our interpretable sparse lexical representations can enhance our understanding of motion, facilitating the creation of more accurate and detailed text annotations. This would improve the quality of datasets and advance human motion understanding and generation tasks.

---

> > ### Comment · Reviewer_uh5q · 2024-11-24
> >
> > Thanks for the response. Although the quantitative results show the superiority over the baseline methods, the Recall results are quite poor.  Does that mean that the learned lexical representation cannot generalize well for different motion distributions?

---

> ### Author Response · Authors · 2024-11-19
> **Response to uh5q (2/2)**
>
> > **W2: Interpretability of motion representation: From the visualization results in Figure 5, it seems that there exist some meaningless words in the word cloud visualization result. Does that mean that the representation is not compact or noisy?**
>
> **A:** We sincerely thank the reviewer for this insightful comment, which has highlighted critical challenges and provided valuable guidance for future improvements to our approach.  As shown in Figure 5, the visualizations demonstrate that the learned lexical representations align well with motion semantics, capturing key terms such as “monkey,” “kick,” “flail,” and “dance waltz,” underscoring the effectiveness of our method. As indicated in Figure 4, our method typically activates approximately 128 keywords, which is significantly more compact compared to current methods that rely on 256 or 512 dimensions. Furthermore, we can enhance regularization constraints to increase the model's focus on fewer, more relevant keywords, thus improving the compactness of the representations.
>
> The activation of seemingly unrelated keywords in lexical representation can be attributed to several factors:
>
> 1. Semantic Relationships Between Words:
> Lexical representation captures the complex semantic relationships between words. As a result, some activated words, though not explicitly present in the caption, may hold implicit relevance to the motion context. For instance, in a "dance waltz" motion, the word "partner" might be activated due to its strong association with waltzing, even if it is not directly mentioned in the caption.
> 2. Diversity and Subjectivity in Data Annotation:
> On average, each motion receives 3.0 distinct textual annotations. The same motion may be annotated from different perspectives in the dataset, leading to variations in the learned representation. For example, while one annotation may focus on the action, another may describe the setting or character. This diversity enriches the representation but may also introduce keywords that seem unrelated at first glance.
> 3. Balancing Regularization and Context Capture:
> While regularization ensures a sparse and compact representation, overly aggressive constraints could suppress important contextual information. The inclusion of these seemingly unrelated words reflects the model’s attempt to balance sparsity with semantic richness.
> 4. Dataset and Task-Specific Factors:
> Some unrelated activations may also stem from inherent limitations in the dataset, such as inconsistent captioning styles or imbalanced keyword distributions. For example, frequent words like "person" or "the" may dominate due to dataset biases.
>
> Overall, these activations are not inherently a drawback but rather a reflection of the lexical representation’s ability to capture both explicit and implicit semantic relationships. As the first to introduce lexical representation to the motion-language domain, our study represents an initial exploration in this direction. Future work could aim to refine the alignment between activated words and task-specific semantics, further enhancing the compactness and relevance of the representation.
>
> We sincerely thank the reviewer for this insightful comment, which has highlighted critical challenges and provided valuable guidance for future improvements to our approach.
>
> ----
>
> > **W3: In Figure 2, the caption of the four modules is out of order.**
>
> **A:** We correct this presentation as follows:
>
> _The framework of our method, including i) LexMLM, which enhances the language model's focus on high-entropy motion-related words; ii) CMMM, which captures spatial and temporal dynamics for robust motion representation; iii) LexMMM, which enables the motion model to identify semantic features and improve cross-modal understanding; and iv) LexCMLP, which aligns motion and text within a unified vocabulary space, ensuring cross-modal coherence._
>
> We will double-check the manuscript and perfect the final version.

---

> ### Author Response · Authors · 2024-11-24
>
> We sincerely thank the reviewer for their insightful comment regarding the Recall results in motion-text retrieval.  This is an important issue in the motion-language domain, and we would like to clarify the underlying challenges while highlighting our method's contributions.
>
> ### **1. Dataset Quality and Scale**
> Motion-text datasets face significant limitations in both quality and scale:
>
> + **High Caption Similarity**: Captions in HumanML3D are often simple, such as _"a person action,"_ leading to high textual similarity (e.g., 0.71 compared to 0.56 in image-text tasks`[1]`). This hampers the discriminative power of textual queries, making text-motion retrieval challenging.
> + **Limited Dataset Size**: Datasets like HumanML3D (14k motion-text pairs) and Motion-X (81k pairs) are considerably smaller and less diverse compared to datasets in other domains. For instance, CLIP leverages 400M image-text pairs, achieving Recall@1 (R@1) scores exceeding 80 or even 90 in image-text retrieval tasks.
>
> These constraints naturally result in lower Recall scores for motion-text retrieval. Nonetheless, our method achieves an R@1 of 11.8, surpassing baselines such as TEMOS (2.12), T2M (1.80), TMR(8.92)  and MotionPatch (10.8). As shown in Figures 5 and 6, our approach activates relevant keywords beyond the simplistic captions, effectively addressing the challenges posed by simple annotations.
>
> ### **2. Cross-Dataset Generalization**
> Generalizing across datasets is particularly challenging due to substantial differences in motion characteristics, including sequence lengths and scenarios. For instance, HumanML3D contains simpler motion contexts and shorter text descriptions (average length: 12 words), whereas Motion-X features more diverse and complex annotations (average length: 38 words).
>
> Such discrepancies make it difficult for existing methods to adapt across datasets, as reflected in their Recall scores. However, our approach, which emphasizes keyword-level importance, demonstrates superior cross-dataset performance compared to baselines, highlighting its robustness and enhanced generalization capability.
>
> ### **3. Future Directions**
> To address these challenges, we plan to improve dataset annotation quality in the future. As demonstrated in Figures 5 and 6, our method identifies additional relevant keywords beyond the dataset's simplistic captions. These insights can guide the enrichment and diversification of annotations, improving dataset quality and enabling finer-grained motion-text retrieval.
>
> Additionally, investigating distributional differences between datasets and developing strategies to enhance cross-dataset generalization are promising directions for future work.
>
> ### **4. Conclusion**
> While limitations in dataset quality and scale contribute to lower Recall scores compared to other domains, our method demonstrates clear improvements over existing approaches across various understanding and generation tasks, including text-motion retrieval, motion captioning, text-to-motion generation, and cross-dataset generalization, thereby validating its effectiveness and generalization capabilities.
>
>
>
> **Reference**
>
> `[1]`Petrovich M, Black M J, Varol G. TMR: Text-to-motion retrieval using contrastive 3D human motion synthesis[C]//Proceedings of the IEEE/CVF International Conference on Computer Vision. 2023: 9488-9497.

---

### Official Review · Reviewer_Ciap · 2024-11-05

**Soundness:** 3
**Presentation:** 3
**Contribution:** 3
**Rating:** 8
**Confidence:** 4

**Summary:**

The main idea of the paper is to address the limitations of existing methods in comprehending human motion by proposing a new motion-language representation paradigm. It introduces a novel method aimed at enhancing the interpretability of motion representations, where a universal motion-language space is proposed to lexicalize and align both motion and text features. In practice,  a multi-phase training strategy, which includes various modeling schemes, is proposed to optimize the motion-language space. Through comprehensive analysis and extensive experiments across multiple public datasets, the proposed model demonstrates state-of-the-art performance in various tasks and scenarios.

**Strengths:**

1. The proposed representation uses a unified motion-language space, effectively capturing and aligning complex motion and textual information. It allows for an efficient and semantically rich reconstruction of human motions, as demonstrated through experimental results.

2. The model design ensures that both motion and text features are concretely lexicalized, providing a clear and interpretable mapping between the elements of the features and their specific semantic meanings

3. A unified multi-phase training strategy is proposed to optimize the motion-language space.

4. The paper is well-written and presents its main ideas, making it accessible to readers and facilitating a deep understanding of the proposed methodology and its applications to motion understanding.

**Weaknesses:**

1. Some ablation studies about the motivation behind the proposed lexical representation are still required. The authors emphasize that some existing language representations are not effective in aligning semantic keywords essential for human comprehension. Is this related to the representation ability of the text encoder? If a more powerful text encoder, such as T5-XXL, were used, would the design of a lexical representation still be necessary? Verifying that even a strong text encoder struggles to align with certain motions would be crucial for validating the motivation behind the proposed scheme.

2. Some implementation details are still required. For example, it is better to provide more detailed descriptions of how the Lexical Disentanglement Head transforms dense motion and text embeddings into sparse lexical representations, rather than just citing a reference. This additional information would help in better understanding and replicating the proposed method.

3. Although the authors have already validated the effectiveness of the proposed representation learning method on retrieval and captioning tasks, I still think it is essential to design experiments to verify its performance on text2motion generation tasks. As the input of text2motion, language can be more flexible and imaginative. If the motion generation can produce reasonable motions from flexible and creative language descriptions, such as "wave the hands and jump in an S-shaped path," it would provide stronger evidence for the effectiveness of the lexical vocabulary space.

**Questions:**

1. From the motion captioning results presented in Appendix A.0.4, the generated captions appear to be nearly identical to the GT, raising concerns about potential overfitting. It would be beneficial if the authors could test more zero-shot examples to further validate the model's generalization capabilities. For instance, taking a sample from a different dataset, such as Motion-X[a], and evaluating the model's performance could provide additional insights into its robustness and versatility.

2. In line 81, the connotation of this LexMLM appears to be incorrectly stated and is inconsistent with the content in the abstract.

[a] Motion-X: A Large-scale 3D Expressive Whole-body Human Motion Dataset

**Details Of Ethics Concerns:**

All of the experiments focus on standard computer vision tasks, and therefore, I believe no ethical review is required.

---

> ### Author Response · Authors · 2024-11-19
> **Response to Ciap (1/3)**
>
> _Thank you for your constructive and thoughtful comments. They were indeed helpful in improving the paper. We take this opportunity to address your concerns:_
>
> _**List of changes in the manuscript:**_
>
> > 1. `Line 81` We have fixed the typo.
> > 2. `Section C.0.1 in Appendix` More ablation studies about the Motivation.
> > 3. `Section C.0.2 in Appendix` Extension to Text-to-Motion Generation.
> > 4. `Section C.0.4 in Appendix` Robustness and Effectiveness Analysis of the Learned Representation.
>
> ---
>
> > **W1:  More ablation studies about the motivation.**
>
> **A:** We appreciate the reviewer for suggesting additional experiments to strengthen our paper.  To further clarify the motivation behind our method, we conducted an ablation study on the state-of-the-art approach, MotionPatch`[1]`, using a series of text encoders of varying model sizes, ranging from T5-Small to T5-XXL. The experimental results are summarized as follows. It is evident that increasing the model size of T5 leads to improved performance. However, despite its significantly larger size, the performance of the T5 model remains substantially inferior to that of the original text encoder, DistilBERT. This indicates that employing a more powerful text encoder does not necessarily enhance the alignment of semantic keywords critical for human understanding.
>
> Prior studies support these observations. For example, in the ablation study of MotionPatch `[1]`, a more advanced text encoder failed to deliver improved representational performance.  Similarly, TMR`[2]` advised the use of DistilBERT instead of the larger CLIP series text encoders.   Additionally, TMR++`[3]` analyzed text annotations across various datasets and demonstrated that while text augmentations can partially bridge the domain gap, significant differences persist between datasets.
>
> Hence, simply adopting a more powerful text encoder does not address the core issue identified in our study. A deeper and more nuanced understanding of the correlation between textual descriptions and motion is required. **Consequently, the design of a lexical representation in our paper is both necessary and justified.**
>
> ---
>
> | Text Encoder | Parameters |  | Text |  to | Motion |   Retrieval | | | Motion | to  | Text |    Retrieval |  |
> | --- | --- | --- | --- | --- | --- | --- | --- | --- | --- | --- | --- | --- | --- |
> |  |  | R@1 $ \uparrow $ | R@2 $ \uparrow $ | R@3 $ \uparrow $ | R@5 $ \uparrow $ | R@10 $ \uparrow $ | MedR $ \downarrow $ | R@1 $ \uparrow $ | R@2 $ \uparrow $ | R@3 $ \uparrow $ | R@5 $ \uparrow $ | R@10 $ \uparrow $ | MedR $ \downarrow $ |
> | T5-Small | 80M | 4.40 | 7.14 | 10.02 | 14.19 | 21.25 | 60.00 | 5.66 | 6.66 | 9.88 | 13.85 | 19.79 | 68.00 |
> | T5-Base | 250M | 4.95 | 7.62 | 10.31 | 14.81 | 23.62 | 46.00 | 5.59 | 6.98 | 10.59 | 14.49 | 20.99 | 54.00 |
> | T5-Large | 780M | 5.82 | 8.72 | 11.66 | 16.93 | 26.27 | 37.00 | 6.69 | 8.33 | 12.16 | 16.70 | 23.53 | 45.00 |
> | T5-XL | 3B | 7.39 | 11.00 | 14.65 | 20.65 | 31.29 | 28.00 | 7.94 | 10.06 | 15.08 | 19.88 | 29.05 | 33.00 |
> | T5-XXL | 11B | 8.41 | 12.82 | 15.96 | 23.67 | 34.42 | 24.00 | 8.97 | 11.49 | 16.69 | 22.63 | 32.16 | 28.00 |
> | DistilBERT | 66M | 10.80 | 14.98 | 20.00 | 26.72 | 38.02 | 19.00 | 11.25 | 13.86 | 19.98 | 26.86 | 37.40 | 20.50 |
> ---
>
> > **W2: More detailed Implementation.**
>
> **A:** We will provide a more detailed description in the final version, including dimensional transformations and processing steps.
>
> Specifically, we use a text encoder to obtain dense text embeddings, denoted as $ E_t \in \mathbb{R}^{L \times D} $, where $ L $ represents the sequence length and $ D $ the embedding dimension. These dense embeddings are then mapped to the lexical space $ E_{\text{lex}} \in \mathbb{R}^{L \times V} $ through an LM head, where $ V $ is the vocabulary size. This process is expressed in Eq. (1).
>
> Next, we transform $ E_{\text{lex}} $ into a global representation by applying max pooling, capturing the maximum value for each lexical token. As a result, the dimensions change from $ L \times V $ to $ V $, where each dimension represents the importance of a corresponding lexical token. To ensure non-negativity and introduce sparsity, we apply a ReLU activation function. Additionally, to mitigate the influence of outliers and reduce the dominance of extreme values, we use a logarithmic saturation function. Finally, to further control the activation of irrelevant tokens, we employ FLOPs regularizer to encourage sparsity and focus the model on more significant tokens. Through these steps, we obtain a sparse lexical representation.
>
> The same process is applied to the motion data. First, dense embeddings are extracted via a motion encoder. These embeddings are then mapped to the lexical space $ E_{\text{lex}} $ using an LM head, followed by max pooling to obtain a global representation. Finally, we apply ReLU, the logarithmic function, and the FLOPs regularizer to ensure sparsity and stability in the representation.

---

> ### Author Response · Authors · 2024-11-19
> **Response to Ciap (2/3)**
>
> > **W3:  Extension to text-to-motion generation task.**
>
> **A:** Our method can be seamlessly integrated into text-to-motion (T2M) generation tasks. Specifically, we directly replace the text encoder in existing methods with our proposed lexical text encoder to extract a more refined and sparse lexical representation. We observe consistent performance improvements across various generative architectures by integrating our method.
>
> Comparative experiments were conducted on multiple frameworks, including VAE-based methods (e.g., T2M`[6]`), diffusion-based approaches (e.g., MDM`[7]`), autoregressive models (e.g., T2M-GPT`[8]`),  and non-autoregressive models (e.g., MoMask`[9]`).
>
> The results, summarized in the following tables, show notable gains, particularly in metrics evaluating motion-text alignment, such as R-precision (Top-1, Top-2, Top-3 accuracy) and Multimodal Distance (MM-Dist). These improvements are primarily attributed to two factors: (1) CLIP-based text encoders, pre-trained on image-text datasets, do not sufficiently focus on motion-related content, and (2) these encoders fail to capture temporal information during the text-image pretraining process. In contrast, our text encoder is specifically designed to identify motion-relevant keywords during pretraining and align them with a motion encoder that captures spatiotemporal motion information, leading to enhanced performance in T2M tasks. This aligns with recent observations in specialized T2M studies, such as HumanTomato`[4]` and LGTM`[5]`, which highlight similar limitations of original CLIP text encoders. However, the majority of mainstream T2M research still emphasizes improving generation paradigms to achieve better results.
>
> Similarly, the importance of effective conditional encoders is being increasingly recognized in other domains, such as text-to-image generation and music-to-dance synthesis.
>
> In summary, simply substituting our method for the original encoder provides measurable performance enhancements. This is an initial attempt, and as the field of human motion research continues to evolve, we aim to explore its application to a broader range of human motion-related tasks.
>
> | **Dataset** | **Paradigms** | **Methods** | **FID ↓** | **Top1 ↑** | **Top2 ↑** | **Top3 ↑** | **MM-Dist ↓** |
> | --- | --- | --- | --- | --- | --- | --- | --- |
> | **HumanML3D** | **VAE** | T2M`[6]` | $ 1.087^{\pm .021} $ | $ 0.455^{\pm .003} $ | $ 0.636^{\pm .003} $ | $ 0.736^{\pm .002} $ | $ 3.347^{\pm .008} $ |
> |  |  | **T2M**$ ^{\S} $ | $ 0.942^{\pm .009} $ | $ 0.472^{\pm .004} $ | $ 0.653^{\pm .002} $ | $ 0.748^{\pm .003} $ | $ 3.104^{\pm .006} $ |
> |  | **Diffusion** | MDM`[7]` | $ 0.544^{\pm .044} $ | $ 0.320^{\pm .005} $ | $ 0.498^{\pm .004} $ | $ 0.611^{\pm .007} $ | $ 5.566^{\pm .027} $ |
> |  |  | **MDM**$ ^{\S} $ | $ 0.524^{\pm .036} $ | $ 0.357^{\pm .004} $ | $ 0.536^{\pm .003} $ | $ 0.643^{\pm .005} $ | $ 5.212^{\pm .021} $ |
> |  | **AR** | T2M-GPT`[8]` | $ 0.141^{\pm .005} $ | $ 0.492^{\pm .003} $ | $ 0.679^{\pm .002} $ | $ 0.775^{\pm .002} $ | $ 3.121^{\pm .009} $ |
> |  |  | **T2M-GPT**$ ^{\S} $ | $ 0.133^{\pm .005} $ | $ 0.506^{\pm .004} $ | $ 0.684^{\pm .003} $ | $ 0.781^{\pm .004} $ | $ 3.002^{\pm .006} $ |
> |  | **NAR** | MoMask`[9]` | $ 0.045^{\pm .002} $ | $ 0.521^{\pm .002} $ | $ 0.713^{\pm .002} $ | $ 0.807^{\pm .002} $ | $ 2.958^{\pm .008} $ |
> |  |  | **MoMask**$ ^{\S} $ | $ \mathbf{0.041}^{\pm .002} $ | $ \mathbf{0.532}^{\pm .002} $ | $ \mathbf{0.721}^{\pm .003} $ | $ \mathbf{0.814}^{\pm .002} $ | $ \mathbf{2.852}^{\pm .008} $ |
>
> | **Dataset** | **Paradigms** | **Methods** | **FID ↓** | **Top1 ↑** | **Top2 ↑** | **Top3 ↑** | **MM-Dist ↓** |
> | --- | --- | --- | --- | --- | --- | --- | --- |
> | **KIT-ML** | **VAE** | T2M`[6]` | $ 3.022^{\pm .107} $ | $ 0.361^{\pm .005} $ | $ 0.559^{\pm .007} $ | $ 0.681^{\pm .007} $ | $ 3.488^{\pm .028} $ |
> |  |  | **T2M**$ ^{\S} $ | $ 2.836^{\pm .062} $ | $ 0.372^{\pm .004} $ | $ 0.574^{\pm .004} $ | $ 0.695^{\pm .005} $ | $ 3.235^{\pm .016} $ |
> |  | **Diffusion** | MDM`[7]` | $ 0.497^{\pm .021} $ | $ 0.164^{\pm .004} $ | $ 0.291^{\pm .004} $ | $ 0.396^{\pm .004} $ | $ 9.191^{\pm .022} $ |
> |  |  | **MDM**$ ^{\S} $ | $ 0.482^{\pm .009} $ | $ 0.214^{\pm .005} $ | $ 0.319^{\pm .004} $ | $ 0.418^{\pm .005} $ | $ 8.682^{\pm .014} $ |
> |  | **AR** | T2M-GPT`[8]` | $ 0.514^{\pm .029} $ | $ 0.416^{\pm .006} $ | $ 0.627^{\pm .006} $ | $ 0.745^{\pm .006} $ | $ 3.007^{\pm .023} $ |
> |  |  | **T2M-GPT**$ ^{\S} $ | $ 0.502^{\pm .016} $ | $ 0.423^{\pm .005} $ | $ 0.641^{\pm .006} $ | $ 0.752^{\pm .006} $ | $ 2.927^{\pm .015} $ |
> |  | **NAR** | MoMask`[9]` | $ 0.204^{\pm .011} $ | $ 0.433^{\pm .007} $ | $ 0.656^{\pm .005} $ | $ 0.781^{\pm .005} $ | $ 2.779^{\pm .022} $ |
> |  |  | **MoMask**$ ^{\S} $ | $ \mathbf{0.186}^{\pm .009} $ | $ \mathbf{0.441}^{\pm .006} $ | $ \mathbf{0.668}^{\pm .004} $ | $ \mathbf{0.792}^{\pm .005} $ | $ \mathbf{2.693}^{\pm .013} $ |

---

> ### Author Response · Authors · 2024-11-19
> **Response to Ciap (3/3)**
>
> > **Q1: More ablation studies about the generalization ability of the proposed method.**
>
> **A:** We appreciate the reviewer for suggesting additional experiments to strengthen our paper.  We conduct the zero-shot test on the Motion-X dataset. Specifically, we utilize the proposed model trained on the Humanml3D dataset to test the samples of the Motion-X dataset on the motion caption task for further validation of the model's generalization capabilities.
>
> We can see that our method can also achieve better performance than the existing methods, which can effectively demonstrate the good generalization capability of our method.
>
> | \Methods |  |  Text to  |Motion | Retrieval   |  |  | |   Motion | to Text | Retrieval |  |  |
> | --- | --- | --- | --- | --- | --- | --- | --- | --- | --- | --- | --- | --- |
> |  | R@1 $ \uparrow $ | R@2 $ \uparrow $ | R@3 $ \uparrow $ | R@5 $ \uparrow $ | R@10 $ \uparrow $ | MedR $ \downarrow $ | R@1 $ \uparrow $ | R@2 $ \uparrow $ | R@3 $ \uparrow $ | R@5 $ \uparrow $ | R@10 $ \uparrow $ | MedR $ \downarrow $ |
> | TMR | 0.06 | 0.11 | 0.24 | 0.39 | 0.51 | 829.00 | 0.08 | 0.16 | 0.21 | 0.37 | 0.56 | 861.50 |
> | MotionPatch | 0.12 | 0.28 | 0.49 | 0.78 | 0.96 | 768.00 | 0.14 | 0.29 | 0.48 | 0.82 | 0.98 | 781.00 |
> | Ours | 0.24 | 0.46 | 0.82 | 0.98 | 1.24 | 503.00 | 0.26 | 0.48 | 0.86 | 1.02 | 1.43 | 516.00 |
>
>
> ----
>
> > **Q2: Typos in Line 81.**
>
> **A:** We correct this presentation as follows:
>
> _Lexical Bottlenecked Masked Language Modeling (LexMLM), which enhances the pre-trained language model’s focus on high-entropy motion-related words for capturing the motion semantics._
>
>
> We will double-check the manuscript and perfect the final version.
>
> ----
>
> **Reference**
>
> `[1]`Yu Q, Tanaka M, Fujiwara K. Exploring Vision Transformers for 3D Human Motion-Language Models with Motion Patches[C]//Proceedings of the IEEE/CVF Conference on Computer Vision and Pattern Recognition. 2024: 937-946.
>
> `[2]`Petrovich M, Black M J, Varol G. TMR: Text-to-motion retrieval using contrastive 3D human motion synthesis[C]//Proceedings of the IEEE/CVF International Conference on Computer Vision. 2023: 9488-9497.
>
> `[3]`Bensabath L, Petrovich M, Varol G. A Cross-Dataset Study for Text-based 3D Human Motion Retrieval[C]//Proceedings of the IEEE/CVF Conference on Computer Vision and Pattern Recognition. 2024: 1932-1940.
>
> `[4]`Lu S, Chen L H, Zeng A, et al. HumanTOMATO: Text-aligned Whole-body Motion Generation[C]//Forty-first International Conference on Machine Learning.
>
> `[5]`Sun H, Zheng R, Huang H, et al. LGTM: Local-to-Global Text-Driven Human Motion Diffusion Model[C]//ACM SIGGRAPH 2024 Conference Papers. 2024: 1-9.
>
> `[6]`Guo C, Zou S, Zuo X, et al. Generating diverse and natural 3d human motions from text[C]//Proceedings of the IEEE/CVF Conference on Computer Vision and Pattern Recognition. 2022: 5152-5161.
>
> `[7]`Guy Tevet, Sigal Raab, Brian Gordon, Yonatan Shafir, Daniel Cohen-Or, and Amit Haim Bermano. Human motion diffusion model. In Proceedings of the International Conference on Learning Representations. OpenReview.net, 2023.
>
> `[8]`Zhang J, Zhang Y, Cun X, et al. Generating human motion from textual descriptions with discrete representations[C]//Proceedings of the IEEE/CVF conference on computer vision and pattern recognition. 2023: 14730-14740.
>
> `[9]`Guo C, Mu Y, Javed M G, et al. Momask: Generative masked modeling of 3d human motions[C]//Proceedings of the IEEE/CVF Conference on Computer Vision and Pattern Recognition. 2024: 1900-1910.

---

> > ### Comment · Reviewer_Ciap · 2024-11-24
> > **Response to Rebuttal**
> >
> > Thanks for providing such a thorough and detailed response. The authors' explanations have effectively addressed all my previous concerns. Therefore, I am pleased to raise my assessment of this paper.

---

> > > ### Author Response · Authors · 2024-11-24
> > >
> > > We sincerely thank you for your thoughtful feedback and kind words of encouragement. We greatly appreciate your recognition of our efforts to address your concerns and your positive reassessment of our work. Your insightful comments have been invaluable in refining our paper, and we are truly grateful for your support and constructive guidance throughout this process.

---

### Author Response · Authors · 2024-11-19
**Overall Response**

We would like to thank all of the reviewers for their constructive and valuable feedback on our work!

**In this post:**

+ (1) We furthermore summarize the strengths of our paper from the reviewers.
+ (2) We summarize the changes to the updated PDF document.

**In the individual replies**, we address other comments.

**- (1) **_**Strengths of Our Paper**_** -**

+ **Sound Motivation**
  - `jkCN`: "The motivation of this paper is sound and non-trivial, the alignment of motion and text is vital for better joint representation learning of human motion and language."
  - `uh5q`: "Introducing the lexical representation paradigm to the text-motion domain is novel."
+ **Novelty of our proposed method**
  - `DGmS`:  "The paper introduces a pioneering approach to human motion-language understanding by creating a unified motion-language space that enhances interpretability through lexical representation. "
  - `Ciap`:  "The model design ensures that both motion and text features are concretely lexicalized, providing a clear and interpretable mapping between the elements of the features and their specific semantic meanings."
  - `uh5q`: Introducing the lexical representation paradigm to the text-motion domain is novel.
+ **Good Presentation**
  - `Ciap`: The paper is well-written and presents its main ideas, making it accessible to readers and facilitating a deep understanding of the proposed methodology and its applications to motion understanding.
  - `DGmS`: The paper is well-organized and easy to follow.
  - `uh5q`: The paper is well written and easy to follow.
+ **Solid Experiments**
  - `jkCN`: The adopted methods make sense and show decent performance on widely used benchmarks and tasks.
  - `uh5q`: Experimental results show remarkable improvements over baseline methods.
  - `DGmS`: The authors have conducted extensive experiments and provided thorough analyses. The model achieves state-of-the-art results across multiple benchmarks, indicating the effectiveness of the proposed approach.
+ **Clear Interpretability & Visualization**
  - `jkCN`: Fig 5 shows well visualization.
  - `DGmS`: The proposed method enjoys better interpretability compared to existing method, which is further validated by the provided qualitative results.



**- (2) **_**Change to PDF**_** -**

We have proofread the paper and added extra experimental results in the revised version (highlighted in blue).

**Main text**

Only small fixes and wording improvements:

+ `Ciap`: (Line 81) We have fixed the typo.
+ `uh5q`: (Figure 2) We have fixed the order.
+ `jkCN`: (Figure 1) We redrew the figure as requested and aligned the relevant content.
+ `jkCN`: (Figure 2) We provide a more detailed caption.
+ `jkCN`: (Figure 3)  We have increased the font size.
+ `jkCN`: (Figure 6)  We have fixed it.
+ `jkCN`: (Figure 7)  We have increased the font size.
+ `jkCN`: (Figure 8) We have updated the caption. The red fonts mean the keywords.
+ `jkCN`: (Line 148) We have fixed the typo.
+ `jkCN`: (Line 192) We have fixed the typo.

**Appendix**

Additional experiments and analyses have been incorporated in response to the reviewers' suggestions:

+ `Ciap, DGmS`: (Section C.0.2)  Extension to Text-to-Motion Generation.
+ `Ciap, uh5q`: (Section C.0.4) Robustness and Effectiveness Analysis of the Learned Representation.
+ `Ciap`: (Section C.0.1) More Ablation Studies about the Motivation.
+ `DGmS`: (Section C.0.3) Complexity Analysis of the Model

---

### Meta-Review · Area_Chair_9Q5Y · 2024-12-16

**Metareview:**

All the 4 reviewers provide positive ratings after rebuttal, with 1 upgraded score. Initially, the reviewers had concerns about ablation studies, implementation details, more results like text-to-motion and generalization. In the post-rebuttal discussion period, all the reviewers are satisfactory with the authors' comments and revised paper. After taking a close look at the paper, rebuttal, and discussions, the AC agrees with reviewers' feedback of the proposed method being sound and effective to solve a challenging task for mapping between motion and language representations. Therefore, the AC recommends the acceptance rating.

**Additional Comments On Reviewer Discussion:**

Initially, the reviewer Ciap had several comments about ablation studies and text-to-motion generation (also raised from reviewer DGmS). Moreover, the reviewer uh5q had questions on the training dataset size with respect to the generalization issue and interpretability of motion representations. After the rebuttal, the authors have provided more explanations and results while updating the paper actively, in which all the reviewers are satisfactory with the answers. The AC considers all the feedback and agrees with reviewers' assessment.

---

### Decision · Program_Chairs · 2025-01-22

Accept (Poster)